# N₂O Emissions from Two Austrian Agricultural Catchments Simulated with an N₂O Submodule Developed for the SWAT Model

Cong Wang [1],*, Christoph Schürz [1], Ottavia Zoboli [2], Matthias Zessner [2], Karsten Schulz [1], Andrea Watzinger [3], Gernot Bodner [4] and Bano Mehdi-Schulz [1,4]

1    Institute for Hydrology and Water Management, Department of Water, Atmosphere and Environment, University of Natural Resources & Life Science, 1190 Vienna, Austria; christoph.schuerz@boku.ac.at (C.S.); karsten.schulz@boku.ac.at (K.S.); bano.mehdi@boku.ac.at (B.M.-S.)
2    Institute for Water Quality and Resource Management, TU Wien, 1040 Vienna, Austria; ozoboli@iwag.tuwien.ac.at (O.Z.); mzessner@iwag.tuwien.ac.at (M.Z.)
3    Institute of Soil Research, Department of Forest and Soil Sciences, University of Natural Resources & Life Science, 3430 Tulln an der Donau, Austria; andrea.watzinger@boku.ac.at
4    Institute of Agronomy, Department of Crop Science, University of Natural Resources & Life Science, 3430 Tulln an der Donau, Austria; gernot.bodner@boku.ac.at
*    Correspondence: cong.wang@boku.ac.at

**Abstract:** Nitrous oxide (N₂O) is a potent greenhouse gas stemming mainly from nitrogen (N)-fertilizer application. It is challenging to quantify N₂O emissions from agroecosystems because of the dearth of measured data and high spatial variability of the emissions. The eco-hydrological model SWAT (Soil and Water Assessment Tool) simulates hydrological processes and N fluxes in a catchment. However, the routine for simulating N₂O emissions is still missing in the SWAT model. A submodule was developed based on the outputs of the SWAT model to partition N₂O from the simulated nitrification by applying a coefficient ($K_2$) and also to isolate N₂O from the simulated denitrification (N₂O + N₂) with a modified semi-empirical equation. The submodule was applied to quantify N₂O emissions and N₂O emission factors from selected crops in two agricultural catchments by using NH₄NO₃ fertilizer and the combination of organic N and NO₃⁻ fertilizer as N input data. The setup with the combination of organic N and NO₃⁻ fertilizer simulated lower N₂O emissions than the setup with NH₄NO₃ fertilizer. When the water balance was simulated well (absolute percentage error <11%), the impact of N fertilizer application on the simulated N₂O emissions was captured. More research to test the submodule with measured data is needed.

**Keywords:** SWAT; N₂O submodule; N₂O emission factor; agricultural catchments; model performance

## 1. Introduction

Nitrous oxide (N₂O) is a potent greenhouse gas (GHG) that contributes to global warming and mainly stems from agricultural soils [1]. Agricultural activities are responsible for an estimated two-thirds of the total anthropogenic N₂O emissions in the world [2].

To estimate the amount of N₂O emitted from an agricultural field in the absence of measured data, the Intergovernmental Panel on Climate Change (IPCC) recommends estimating the N₂O emissions as a fraction of the N input to the soil. This is known as the N₂O emission factor (EF), which is defined as kg N₂O-N kg⁻¹ N input [3–5]. Initially, calculating the EF for N₂O was based on a simple regression model established by Bouwman [6] who estimated the contribution of fertilizers encompassed 90% of the total N₂O emissions. The IPCC thus proposed a unique EF for all agricultural soils, regardless of variations in soil management and climate [7]. An EF for N₂O of 1.25 ± 1.0% of the applied N fertilizer amount was adopted in 1996 [3,8].

In 2006, after the data from 1215 measurements were analyzed by Stehfest and Bouwman [9], the IPCC revised the EF value of $N_2O$ and derived a value calculated as 0.01 kg $N_2O$-N per kg N applied to a field (uncertainty range 0.003–0.03), which was valid for all fertilizer and manure types, application techniques, and land uses [4,10].

In 2019, based on a much larger number of measurements (~3000) [5], the IPCC refined the 2006 aggregated EF value of $N_2O$, which is now deemed to be 1% of the total applied N fertilizer amount, with an updated uncertainty range of 0.1–1.8%. Furthermore, to highlight the role of different climate conditions on $N_2O$ emissions, the EF can be disaggregated according to wet and dry climatic zones, and in the wet climate zone can then be further disaggregated according to organic and mineral N fertilizer applied. In the dry climate zone, the fertilizer type is not differentiated.

Wet climates are defined to occur in temperate and boreal zones, where the ratio of annual precipitation to potential evapotranspiration is >1, and in tropical zones, where total annual precipitation is >1000 mm [5]. In wet climates, the $N_2O$ EF value is 0.6% of the organic N inputs (uncertainty range of 0.1–1.1%) and is 1.6% of the mineral N inputs (uncertainty range of 1.3–1.9%) [5]. In dry climates, the $N_2O$ EF value is 0.5% of the total organic and mineral N inputs (uncertainty range of 0–1.1%) [5].

In the absence of measured data, process-based simulation models offer an option to capture $N_2O$ emissions from agricultural fields under different management practices at different time steps (e.g., daily, monthly or yearly) and several scales (e.g., field, landscape or catchment). The governing processes to simulate $N_2O$ emissions from N fertilizers are manifold. The reactive N in mineral N fertilizers are ammonium ($NH_4^+$) and nitrate ($NO_3^-$). The $NH_4^+$ part of mineral fertilizers can be oxidized through the nitrification process to $NO_3^-$ by *Nitrosomonas* and *Nitrobacter*, and thereby produce $N_2O$ gas as a by-product [11,12]. The percent of nitrified N lost as $N_2O$ varies with soil type with reported ranges lying anywhere between 0.01% and 95% [13].

Under anaerobic conditions, denitrification reduces the $NO_3^-$, part of mineral fertilizers (or the $NO_3^-$ from the nitrification process), to nitric oxide (NO), then to $N_2O$, and finally to dinitrogen ($N_2$) gas [11]. Thus, $N_2O$ is an intermediate product in the denitrification process. The fraction of global $N_2O$ emitted from denitrification ranges between 0.02 and 0.37 [14].

Nitrification and denitrification are regulated by several field management practices and environmental factors [9,15,16]. The interaction of these factors increase the complexity and uncertainty of assessing $N_2O$ emissions at large spatial scales.

A number of system dynamic models have been developed to simulate $N_2O$ fluxes from agricultural soils. For a detailed description of these as well as their limitations see Wang et al. [12]. For example, the DAYCENT (Daily Century) and the DNDC (DeNitrification-DeComposition) models are two commonly used physically based, biogeochemical models that simulate $N_2O$ emissions at the plot or the field scale [17,18], and the SWAT (Soil and Water Assessment Tool) model can be used to simulate $N_2O$ emissions at the catchment scale [19–25].

The DAYCENT model simulates daily $N_2O$, $N_2$, and $NO_x$ (e.g., NO and $NO_2$) emissions from nitrification and denitrification with the simulated soil water content, temperature, $NH_4^+$, $NO_3^-$, and respiration [18,26]. The DNDC model simulates $N_2O$ emissions by tracking different groups of microbes that are activated under several environmental conditions, including different temperature, moisture, pH, redox potential, and substrate concentration gradients in soils [27].

Somewhat different from these two biogeochemical models is the eco-hydrological SWAT model that simulates hydrological processes and crop growth at a larger (usually the catchment) scale to estimate the impacts of agricultural land use and the corresponding crop management practices on water, atmosphere and soil quality [19,28]. The advantage of using a hydrological model to simulate $N_2O$ emissions is the integration of several field crops in a defined area, and the resulting connectivity of the hydrological flow paths and subsequent N transformations in the area to quantify $N_2O$ emissions. As part of a

larger study, we wished to obtain an integrated estimation of $N_2O$ (and other reactive N) emissions from various crops and their field management practices at the catchment scale. Therefore, the SWAT model was selected as the preferred modelling tool.

However, there are a few challenges for the SWAT model to simulate $N_2O$ emissions. For example, the model does not partition $N_2O$ from nitrification and treats the $N_2O + N_2$ outputs as a combined denitrification product. This grouping of $N_2O + N_2$ limits the ability of the SWAT model to simulate only the $N_2O$ fraction. In SWAT, nitrification is a function of soil temperature and soil moisture, while denitrification is a function of soil moisture, temperature, SOC, and soil $NO_3^-$ [29]. A few studies have been undertaken to develop a SWAT $N_2O$ submodule to partition $N_2O$ emissions from the SWAT simulated nitrification and denitrification [20–25]. All the submodules use different approaches and equations.

For example, Yang et al. [22] integrated all the biogeochemical $N_2O$ emission algorithms from DAYCENT that were developed by Del Grosso et al. [30] and Parton et al. [18,26] into the SWAT model and did not use the SWAT default equations for simulating nitrification and denitrification based on SWAT simulated variables. The resulting SWAT-$N_2O$ model was applied to grain corn, switchgrass and brome grass fields in southwestern Michigan.

Furthermore, Wagena et al. [23] developed the SWAT_GHG model to assess $N_2O$ emissions from agroecosystems in North America. They integrated equations from Parton et al. [26], which were developed based on laboratory data and on data from Weier et al. [31], into the SWAT model to consider the impacts of soil temperature, soil moisture and pH on $N_2O$ emissions from the nitrification process. The SWAT_GHG model adopted equations from Parton et al. [26] to calculate denitrification and the $N_2/N_2O$ ratio, and thereby considered the impacts of soil $NO_3^-$, SOC, and soil moisture. They also developed additional equations to include the impacts of soil pH on $N_2O$ emissions.

Shrestha et al. [24,25] estimated $N_2O$ emissions for catchments in Canada using SWAT by applying all equations from Parton et al. [26] into the SWAT model thereby did not use the initial equations in SWAT. Unlike the other two studies, Shrestha et al. [24,25] did not consider the impact of soil pH on denitrification and the $N_2/N_2O$ ratio, since the SWAT model by default does not account for changes in soil pH. Also in their study, the only mineralized N was in the form of ammonium. However, SWAT adds the nitrogen mineralized from the active organic and fresh organic pools to the soil $NO_3^-$ pool, which leads to $N_2O$ emissions from organic N input through the denitrification process.

Most recently, Gao et al. [20] established a database for $N_2O$ fluxes based on 4488 field measurements, and developed an empirical equation to simulate $N_2O$ emissions from an agricultural catchment in northeast China, specifically to predict $N_2O$ emissions from non-paddy fields. The $N_2O$ emissions from non-paddy fields are directly calculated with the simulated variables from SWAT of soil moisture, soil temperature and soil nitrogen [20].

Although several submodules exist for simulating $N_2O$ in the SWAT framework, they do not have a common agreement to simulate $N_2O$ emissions. The submodules developed by Yang et al. [22] and Wagena et al. [23] included soil pH. However, the changes in soil pH are not captured by the SWAT model. Furthermore, current studies show contradictory results regarding the impact of soil pH on nitrification and denitrification [12]. In addition, Yang et al. [22], Wagena et al. [23], and Shrestha et al. [24] adopted equations developed for the DAYCENT model from Parton et al. [26] to calculate nitrification and denitrification. Hydrological processes such as precipitation, infiltration, and runoff are particularly important in determining $NO_3^-$ transformation and transportation, as well as $N_2O$ emissions. Thus, to simulate $N_2O$ emissions at the catchment scale, the $N_2O$ submodule should be ideally developed based on the variables simulated by a distributed hydrological model (e.g., soil $NO_3^-$, soil moisture, and SOC), such as the SWAT model and should partition $N_2O$ emissions directly from simulated nitrification and denitrification.

According to the estimation of the Environment Agency Austria carried out at the national level, in Austria in 2018, the $N_2O$ emissions from agriculture soils contributed 69% to the total national $N_2O$ emissions [32]. To our knowledge, only a few field studies [33–35]

and modelling exercises [36] were undertaken to quantify $N_2O$ emissions from Austrian cropland at the field scale. Empirical and modelling studies on simulating $N_2O$ emissions at the catchment scale are few and therefore uncertainties in emissions from crop sites remain high.

In this study, we develop an $N_2O$ submodule based on the eco-hydrological SWAT model outputs to quantify the integrated $N_2O$ emissions and $N_2O$ EFs from two agricultural catchments (Melk and Zaya) with two N fertilizer regimes. We analyze the model performance and the impacts of management practices on simulating $N_2O$ emissions by comparing simulated $N_2O$ emissions with existing $N_2O$ data.

## 2. Methodology and Materials

### 2.1. The Case Study Catchments

The Melk and Zaya catchments were initially chosen as part of a large ongoing nationally funded project (NitroClimAT) in which N fluxes into the soil-water-atmosphere were evaluated. Both catchments are representative Austrian agricultural catchments within their respective main production region. These catchments were selected within the project framework to apply the SWAT model with the developed $N_2O$ submodule.

The Melk catchment (282 km$^2$) and the Zaya catchment (522 km$^2$), both of which are located in lower Austria (Figure 1), are two agricultural catchments with typical annual crops for the region. The percentages of cropland and pasture area in the Melk catchment are 63% and 12%, respectively, and in the Zaya catchment are 69% and 8%, respectively. The remaining areas in these two catchments are forest and urban. The main field crops grown in the Melk catchment are winter wheat, grain corn, soybean, and spring barley, while the main crops grown in the Zaya catchment are winter wheat, spring barley, sugar beet, and grain corn (INSPIRE data available from www.inspire.gv.at). Table 1 shows the site characteristics of both catchments.

### 2.2. The SWAT Model and Model Setup

The SWAT model is a process-based model that simulates the impacts of agricultural management practices on the hydrology and water quality most frequently at the catchment scale [19]. In the SWAT model, a catchment is partitioned into a number of subbasins, which are further divided into HRUs (hydrological response units). HRUs are areas that have the same land cover, soil, and slope [19]. When specific crops are examined, these are analyzed at the HRU level, on the various soil and slope types on which they occur. The SWAT model implements routines of the EPIC crop growth submodule to simulate the annual variation in crop growth [29].

A SWAT model setup requires multiple input data, including a DEM (digital elevation model), a soil map (with soil properties), climate data, a land use map, as well as data on field management operations. The link to detailed input data sources and input data preparation for the Melk and Zaya catchments refers to Data Availability Statement.

In this study, the SWAT model in the Melk catchment was set up with two N fertilizer input data (Table 2), while the SWAT model was only set up for the Zaya catchment with the same N fertilization input data as from Melk M2 (Table 2). The M1 is an optimistic setup based on best-management practices and most likely under represents the amount of fertilizer that farmers apply to their fields. The N fertilizer type in M1 that is simulated in the SWAT model is input as $NH_4NO_3$. The amounts and types of applied N fertilizer in M2 are an attempt to represent the actual N fertilization regimes of farmers based on N-balance calculations. The N fertilizer types simulated in M2 are mineral N in the form $NO_3^-$ and organic N.

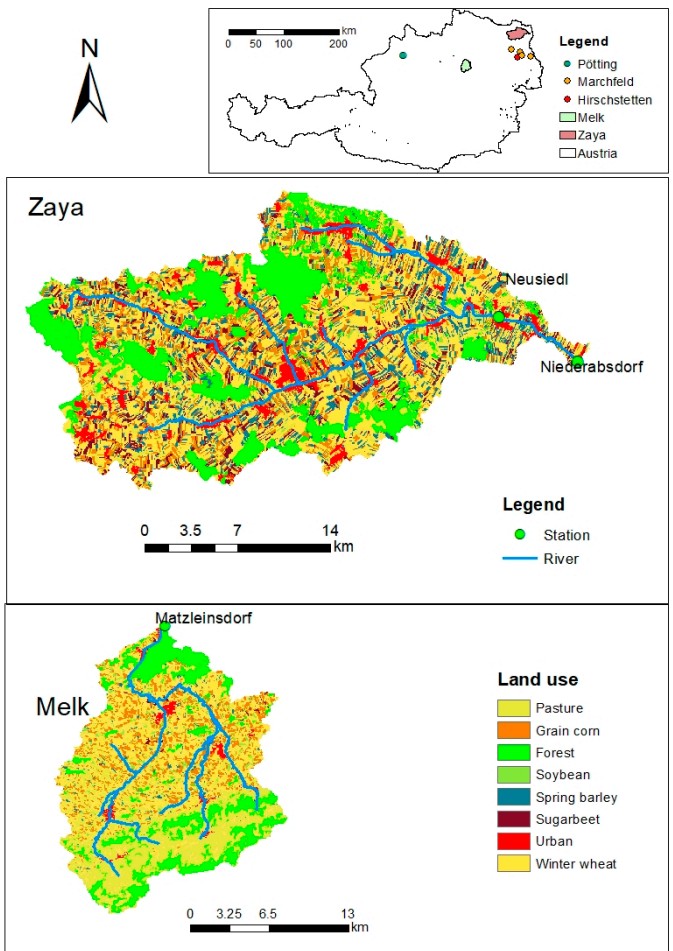

**Figure 1.** The land use maps with crop categories according to SWAT and river distribution in the Melk and Zaya catchments. Matzleinsdorf and Niederabsdorf are the outlet gauges of the Melk catchment and the Zaya catchment, respectively. Pötting, Hirschstetten and Marchfeld are regions where we have measured $N_2O$ data and simulated $N_2O$ data (from the DNDC model), respectively.

**Table 1.** Characteristics of the Melk and Zaya catchments. The soil information is the average value of all soil classes in the first layer (0–300 mm) of the two catchments collected from SoilGrids (https://soilgrids.org) (accessed on 8 August 2018). Daily air temperature, precipitation and other weather data was obtained from ZAMG (https://www.zamg.ac.at/cms/de/aktuell) (accessed on 19 December 2018) and eHYD (https://ehyd.gv.at/) (accessed on 6 August 2018). The resolution of dominant texture is 250 m × 250 m.

| Catchment Characteristics | Unit | Melk | Zaya |
|---|---|---|---|
| Elevation range | m | 201–1050 | 153–490 |
| Mean annual rainfall | mm | 794 | 553 |
| Mean maximum temperature | °C | 14.4 | 13.4 |
| Mean minimum temperature | °C | 5.3 | 5.4 |
| Average clay | % | 23.39 | 28.18 |
| Average silt | % | 44.5 | 49.03 |
| Average sand | % | 32.14 | 22.79 |
| Average rock fragment content | % | 9.81 | 10.33 |
| Average organic carbon content | % | 3.36 | 2.2 |
| Dominant texture | | Silt-loam | Clay loam |
| Average moist bulk density | [g cm$^{-3}$] | 1.32 | 1.34 |
| Average available water content of the soil layer | [mm mm$^{-1}$] | 0.18 | 0.16 |

**Table 2.** Main crops categories, area and the amounts of applied N fertilizer in the Melk and Zaya catchments.

|  | Crop | PAST | WWHT | CORN | SOYB | BARL |
|---|---|---|---|---|---|---|
| M1 | Area (%) | 44.9 | 22.8 | 11.6 | 3.9 | 2 |
|  | Min N (kg ha$^{-1}$) | 140 | 170 | 197.6 | 0 | 130 |
|  | Crop | FESI | WWHT | CORN | SOYB | FESE |
| M2 | Area (%) | 32.1 | 22.8 | 11.6 | 3.9 | 3.6 |
|  | Org N (kg ha$^{-1}$) | 136 | 4.7 | 102 | 4.7 | 10.8 |
|  | Min N (kg ha$^{-1}$) | 35 | 130.1 | 58.3 | 28.2 | 64.8 |
|  | Crop | WWHT | BARL | SGBT | CORN | FESI |
| Zaya | Area (%) | 45.1 | 14.2 | 8.9 | 8.7 | 0.6 |
|  | Org N (kg ha$^{-1}$) | 4.8 | 5.6 | 4.7 | 38.8 | 127 |
|  | Min N (kg ha$^{-1}$) | 139.7 | 92 | 108.9 | 106.2 | 5.9 |

Area (%) is the percent of crop area per agricultural area. Abbreviations: Min N: mineral N, Org N: organic N. The crops included in PAST (pasture), WWHT (winter wheat), CORN (grain corn), SOYB (soybean), BARL (spring barley), SGBT (sugar beet), FESI (intensively managed pasture), and FESE (extensively managed pasture) refer to Tables S2–S4.

The SWAT model simulates N fluxes by several processes (e.g., mineralization, volatilization, nitrification, and denitrification). For example, mineralization converts organic N to $NO_3^-$ [29]. Through nitrification and volatilization, $NH_4^+$ is converted to $NO_3^-$ and $NH_3$, respectively. The SWAT model simulates the total amount of nitrification and $NH_3$ volatilization, and then partitions the nitrification from volatilization [29]. Equations for simulating and partitioning nitrification in the SWAT model are provided in Equations (S1)–(S11).

The SWAT model simulates the amount of $NO_3^-$ lost to denitrification with two equations (Equations (1) and (2)) [29]. These calculate the total gases flux ($N_2O + N_2$) resulting from denitrification:

$$N_{denit} = NO3_{ly} * \left(1 - exp\left(-\beta_{denit} * \gamma_{tmp} * orgC\right)\right) \; if \gamma_{sw,ly} \geq \gamma_{sw,thr}, \tag{1}$$

$$N_{denit} = 0.0 \; if \gamma_{sw,ly} < \gamma_{sw,thr}, \tag{2}$$

where $N_{denit}$ is the amount of nitrogen lost to denitrification per area given in kg N ha$^{-1}$. $NO3_{ly}$ is the amount of $NO_3^-$ per area in kg N ha$^{-1}$ and calculated from the depth of the soil layer, ly. $\beta_{denit}$ is the rate coefficient for denitrification. $\gamma_{tmp}$ is the impact of soil temperature on denitrification. $orgC$ is the amount of organic carbon in the layer (%). $\gamma_{sw,ly}$ is the denitrification water factor for layer ly, and $\gamma_{sw,thr}$ is the threshold value of water factor for denitrification to occur. Equations for the effect of soil temperature and soil water on denitrification are provided in Equations (S12) and (S13).

To check the performance of these existing $N_2O$ submodules, we wrote all equations governing the $N_2O$ emissions from Parton et al. [26], Wagena et al. [23], and Shrestha et al. [24] in the "R" programming language. We then assigned values to each of the environmental factor in the equations to determine their impact on $N_2O$ emissions. The environmental factors include soil $NO_3^-$, SOC, and WFPS, and their ranges plotted were as follows: 0–350 ug N g$^{-1}$, 0–35 kg C ha$^{-1}$ d$^{-1}$, and 0–1, respectively. We thereby determined that a bracket was misplaced in the equation of Parton et al. [26] because it provided negative values when soil $NO_3^-$ concentration in the range of 0–350 ug N g$^{-1}$ were used in the formula to calculate the $N_2/N_2O$ ratio (Figure S1A). We corrected this as shown in Equation (5). Wagena et al. [23] applied the initial equation in their $N_2O$ module as found in Parton et al. [26]; however, Shrestha et al. [24] also corrected Parton et al.'s equation.

We furthermore found that the equation for calculating the impact of soil moisture on the $N_2/N_2O$ ratio in Wagena et al. [23] contained a typo, due to the incorrect position of the exponent (Figure S1B).

### 2.3. The $N_2O$ Submodule

The $N_2O$ submodule was developed in the "R" programming language and was based on using the outputs of nitrification and denitrification from the SWAT model version 2012. The link to related R codes and used R packages refers to Data Availability Statement.

To partition the $N_2O$ from nitrification and from denitrification, we firstly modified the SWAT source code file "nminrl.f" to obtain the simulated nitrification in "output.hru" file, and then partitioned the by-product $N_2O$ from nitrification using a fraction (called $K_2$) of nitrified N lost as $N_2O$. Secondly, the denitrification output of the SWAT model was partitioned using a modified formula from Parton et al. [26] to differentiate $N_2O$ from total gaseous products of denitrification ($N_2O + N_2$). Figure 2 shows the $N_2O$ related processes in the SWAT model and the SWAT $N_2O$ submodule, where the red-dashed line outlines the $N_2O$ submodule was developed for SWAT.

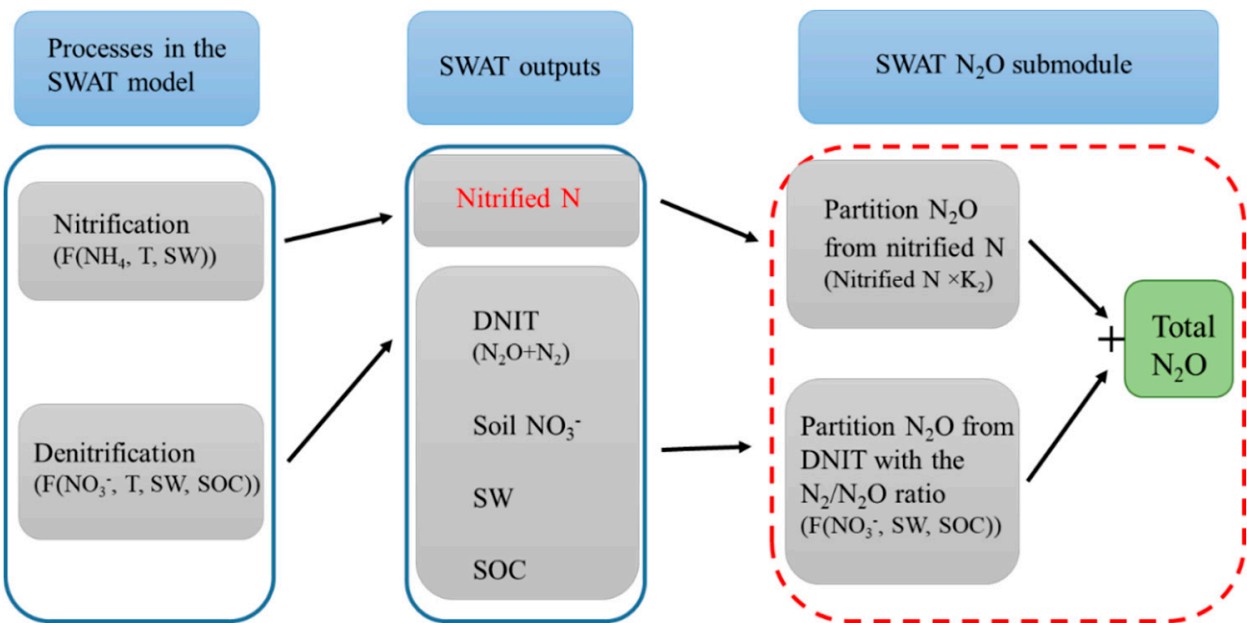

**Figure 2.** Schematic diagram of the processes included in the SWAT model and the SWAT $N_2O$ submodule. Nitrified N in a red color indicates the SWAT model simulates the nitrified N but can only be examined in the output file after the SWAT source codes were modified. Abbreviations: F: function. T: soil temperature. SW: soil water. SOC: soil organic carbon. DNIT: denitrification. $K_2$: the fraction of nitrified N lost as $N_2O$.

The governing equations in the $N_2O$ submodule adopted together with SWAT output to calculate $N_2O$ emissions are provided below.

Partitioning $N_2O$ from the SWAT simulated nitrification is carried out as follows:

$$N_{N_2O} = N_{NO_3} * K_2, \tag{3}$$

where $N_{N_2O}$ is the $N_2O$ flux from nitrification (kg N ha$^{-1}$ d$^{-1}$), $N_{NO_3}$ is the rate of nitrification (kg N ha$^{-1}$ d$^{-1}$), and $K_2$ is the fraction of nitrified N lost as $N_2O$ (%). The $K_2$ value was obtained from the literature for various soil types listed in Table S1. The value of $K_2$ used in the submodule is set as 0.02 based on the results of the extensive literature review (Section 3.3).

For the simulated denitrification process, part of the N$_2$O is converted to N$_2$ under anaerobic conditions. Equation (4) modified from Parton et al. [26] was used to calculate the N$_2$/N$_2$O ratio ($R_{N_2/N_2O}$):

$$R_{N_2/N_2O} = min(F_r(NO_3), F_r(C))F_r(\theta),$$ (4)

where $F_r(NO_3)$, $F_r(C)$, and $F_r(\theta)$ are the environmental impacts of soil NO$_3^-$, soil C, and soil moisture, respectively, on the ratio N$_2$/N$_2$O. We corrected the $F_r(NO_3)$ equation in Parton et al. [26] with the missing bracket. Thus, the final equations for $F_r(NO_3)$, $F_r(C)$, and $F_r(\theta)$ are as follows:

$$F_r(NO_3) = \left(1 - \left(0.5 + \frac{1 * atan(\pi * 0.01 * (NO_3 - 190))}{\pi}\right)\right) * 25,$$ (5)

$$F_r(C) = 13 + \frac{30.78 * atan(\pi * 0.07 * (C - 13))}{\pi},$$ (6)

$$F_r(\theta) = \frac{1.4}{13^{\left(\frac{17}{13^{(2.2*WFPS)}}\right)}},$$ (7)

$$WFPS = \frac{(SW * \rho_b)}{soil\ porosity},$$ (8)

$$soil\ porosity = 1 - \rho_b/2.65,$$ (9)

where NO$_3$ is soil NO$_3^-$ (ug N g$^{-1}$), $C$ is SOC (kg C ha$^{-1}$d$^{-1}$), *WFPS* is water-filled pore space (%), *SW* is soil water content (%), $\rho_b$ is soil bulk density (g cm$^{-3}$).

Therefore, the amount of partitioned N$_2$O from N$_2$ can be calculated as

$$D_{N_2O} = \frac{N_{denit}}{1 + R_{N_2/N_2O}},$$ (10)

where $D_{N_2O}$ is the N$_2$O emissions from the denitrification process (kg N ha$^{-1}$ d$^{-1}$).

In our developed SWAT N$_2$O submodule, the N$_2$O emissions (kg N ha$^{-1}$ d$^{-1}$) from nitrification and denitrification are finally given by

$$N_2O = N_{N_2O} + D_{N_2O}.$$ (11)

### 2.4. Calibrating the SWAT Model

We performed a calibration/validation for the three SWAT model setups (M1, M2, and Zaya). The N$_2$O submodule was not calibrated during this step.

We firstly perform a sensitivity analysis on selected parameters to determine the most sensitive parameters that control the model performance, cause uncertainty, and should be calibrated. The parameters for the sensitivity analysis, their definition, and initial ranges are listed in Table S5.

Simulations with 1000 parameter sets were performed by using the SWATplusR package, which integrates the SWAT model into users' modeling workflows in "R" [37]. The parameter sets were sampled using Latin hypercube sampling within the parameters' ranges (Table S5) [38] using the R package "lhs" (https://CRAN.R-project.org/package=lhs) (accessed on 12 March 2019). The sensitive parameters were selected based on the analysis of cumulative distribution with the PAWN approach [39]. In the PAWN approach, the Kolmogorov–Smirnov statistic is applied to measure the distance between the cumulative probability distributions of an output variable when all parameters are perturbed and when all but the analyzed parameters are perturbed. The performed sensitivity analysis implements the approximation of PAWN that is outlined by Pianosi and Wagener [39].

The SWAT model calibration and validation were performed at a daily time step with a warm-up period of 5 years. For the Melk catchment, the calibration was conducted simultaneously for discharge and NO$_3^-$-N concentration in the M1 setup. In the M2 setup

and for the Zaya catchment, the parameters were calibrated sequentially first for discharge and then for $NO_3^-$-N concentration. The objective function used during the calibration and validation was the Kling-Gupta Efficiency (KGE) [40]. The calibration and validation results were also evaluated by the percentage of bias (PBIAS) [41].

In the Melk catchment, the gauging stations for both discharge and $NO_3^-$-N concentration measurements are located at the outlet, at Matzleinsdorf, and in the Zaya catchment, the gauging stations for measured discharge and $NO_3^-$-N concentration are located at Niederabsdorf and Neusiedl, respectively (Figure 1).

After running the validated SWAT model, we evaluated the model performance for the whole catchment by calculating the water balance for the catchment at the yearly time step, verifying the evapotranspiration (ET) and checking simulated crop yields against measured yields.

The water balance is calculated from the SWAT variables as follows [29]:

$$PREC = Q_{surf} + ET + \Delta SW + PERC - GW\_Q, \tag{12}$$

where PREC is the amount of precipitation (mm); $Q_{surf}$ is the amount of surface runoff (mm); ET is the amount of evapotranspiration (mm); $\Delta SW$ is the change in soil water content (mm); PERC is the amount of water entering the vadose zone from the soil profile (mm) and GW_Q is the amount of groundwater contribution to streamflow (mm). Irrigation was not implemented in the SWAT model for either of the catchments.

The Penman–Monteith equation was selected in SWAT for calculating potential ET [29]. To verify the ET with independent data, the remote sensed ET data from ALEXI (Atmosphere-Land Exchange Inverse) was compared with the simulated ET in the Melk and Zaya catchments at the monthly time step.

The remote sensed ALEXI model with $5 \times 5$ km resolution estimates daily ET data. The ALEXI model is based on a surface energy balance modeling system, and is governed primarily by remote sensing inputs of land surface temperature [42–44]. Cawse-Nicholson et al. [45] quantified the uncertainty of the ALEXI model for estimating ET to have a mean offset of 0.01 mm day$^{-1}$.

The SWAT simulated crop yields for grain corn, winter wheat, spring barley and soybean were compared with measured crop yields in Austria for the years 1993–2018 (Bundesanstalt für Agrarwirtschaft und Bergbauernfragen data available from https://j1 dev.agrarforschung.at/index.php?lang=de) (accessed on 25 April 2020).

*2.5. Evaluating the N₂O Submodule*

Since the two study catchments were chosen as part of a larger on-going project, a thorough research was undertaken to find measured $N_2O$ data in the Melk and Zaya catchments; however, to the best of our knowledge, no measured $N_2O$ data from agricultural field crops was available in these two catchments.

Therefore, to evaluate the submodule performance for simulating $N_2O$ emissions, we compared the simulated $N_2O$ emissions with available measured data in nearby adjacent regions. Measured $N_2O$ data from a proximate site located in Pötting, Upper Austria, located about 140 km away from the outlet gauge in the Melk catchment (Figure 1), was compared with selected crops in the Melk catchment. Pötting is dominated by a silty loam soil type and wet climate that are similar to the input data in the Melk catchment (Table 3). The $N_2O$ data in Pötting was measured during 2013 from a grain corn field at an hourly time step by using a closed chamber method made of PVC cylinders (10 cm height, 20 cm diameter) inserted 3 cm deep into the soil [35].

**Table 3.** Site characteristics of Pötting, Melk, Zaya, Hirschstetten, and MF3.

| Site | Soil Texture | Rainfall (mm) [a] | Crop [b] | N Fertilizer (kg N ha$^{-1}$) | Fertilizer Type | Days in Growing Season | Irrigation |
|---|---|---|---|---|---|---|---|
| Pötting | Silty loam | 891 | Grain corn | 162 | Slurry & Urea | 158 | No |
| M1 | Silty loam | 976 | CORN | 197.6 | $NH_4NO_3$ | 157 | No |
| | | | WWHT | 170 | | 266 | |
| | | | BARL | 130 | | 128 | |
| M2 | | | CORN | 160.3 | $NO_3$ & Organic N | 188 [c] | No |
| | | | WWHT | 134.8 | | 362 [c] | |
| Zaya | Clay loam | 583 | CORN | 145 | $NO_3$ & Organic N | 188 [c] | Real: yes; Model setup: no |
| | | | WWHT | 144.5 | | 291 [c] | |
| | | | BARL | 97.6 | | 133 [c] | |
| | | | SGBT | 113.6 | | 215 [c] | |
| Hirschstetten | Loamy sand | 433 | Spring barley | 50 | $NH_4NO_3$ | 106 | No |
| | | | Winter wheat | 120 | | 265 | |
| MF3 | Loam | 650 | Grain corn | 150 | $NH_4NO_3$ | | Yes |
| | | | Winter wheat | 116 | | | |
| | | | Spring barley | 55 | | | |
| | | | Sugar beet | 110 | | | |

[a]: For Pötting, M1 and M2, rainfall was the amount of total precipitation in 2013. For the other sites, rainfall is the average annual precipitation. [b]: Crops in SWAT (CORN, BARL, and WWHT) and crops with measured $N_2O$ data or simulated in DNDC model (grain corn, spring barley, winter wheat, and sugar beet). [c]: The seeding and harvest dates in SWAT took place on dates without precipitation.

Similarly, measured $N_2O$ data from an outdoor lysimeter experiment in Hirschstetten, Vienna, located about 46 km away from the Zaya outlet gauge (Figure 1), was compared to the simulated $N_2O$ emissions from crops in the Zaya catchment. Hirschstetten has a dry climate similar to the Zaya catchment (Table 3). The measured $N_2O$ data from spring barley and winter wheat in Hirschstetten were collected in 2018 and 2019, respectively.

We furthermore compared the simulated $N_2O$ emissions in the Melk and Zaya catchments with simulated $N_2O$ emissions from the DNDC model used to simulate $N_2O$ emissions from the Marchfeld region (Figure 1) for the years 2006–2011 [36]. From all the sites simulated with DNDC in Kasper et al. [36], we chose the site that had the most similar soil properties with the Melk and Zaya catchments to compare our simulated $N_2O$ emissions. This site is referred to as "MF3" in Kasper et al. [36]. The percentage of clay, silt and sand in Melk (23%, 45%, and 32%, respectively) and Zaya (28%, 49%, and 23%, respectively) are similar to MF3 (28%, 48%, and 24%, respectively). The MF3 site has a dry climate. Wet and dry climate are defined based on the definition in the IPCC report [5].

The similarity of the site characteristics between Pötting and Melk, between Hirschstetten and Zaya, and between MF3 and Melk/Zaya are provided in Table 3.

The measured hourly $N_2O$ emissions in Pötting and Hirschstetten were both aggregated to the daily time step, respectively, to compare with the corresponding simulated daily $N_2O$ emissions from the submodule. Linear interpolation was used to generate the cumulative $N_2O$ flux during the crop growing season [46–48]. In the Zaya catchment for each of the years 2006 to 2015, a one-way analysis of variance (ANOVA) was performed in R to analyze the effect of daily precipitation on daily $N_2O$ emissions during the growing season of spring barley (from approximately 22 March to 1 August each year).

The simulated $N_2O$ emissions from the submodule together with the SWAT applied N fertilizer amounts to the crop categories were used to calculate the EF for $N_2O$ in the Melk and Zaya catchments. The EFs were compared with the aggregated and disaggregated IPCC default EF values for $N_2O$.

## 3. Results

### 3.1. Sensitivity Analysis, Calibration and Validation

The most sensitive parameters for the M1 setup were ALPHA_BF (base flow recession constant), CN2 (curve number), GWQMN (threshold water level in shallow aquifer for base flow), OV_N (Manning's value for overland flow), SNO50COV (fraction of SNOCOVMX that provide 50% cover), CDN (rate coefficient for denitrification), NPERCO (nitrate percolation coefficient), and RCN (concentration of nitrogen in the rain). These most sensitive parameters were further calibrated. For the M2 and Zaya setups, all parameters in Table S5 were calibrated.

The calibration and validation results and time periods for all setups are shown in Table 4. In the M1 and M2 setups, the simulated $NO_3^-$ in the stream are lower than the measured $NO_3^-$ concentrations (Figures S4 and S5) and one main reason is the limited availability of the observed $NO_3^-$-N concentration data, for example, Gungor et al. [49] and Ikenberry et al. [50] highlighted the underestimation of $NO_3^-$. The discharge peaks for the years 1994, 1996, 1997, and 2002 were not well captured (Figure S2), which further affects the simulation of in-stream $NO_3^-$ simulation [51]. As well, the sensitive parameters, such as ALPHA_BF and OV_N in the M1 setup, caused the underestimation of discharge. CDN, NPERCO and RCN are sensitive parameters in the M1 setup that influence the simulation of the mass of $NO_3^-$ in water.

**Table 4.** Model performance for the daily calibration and validation for discharge (Q) and $NO_3^-$-N concentration (N).

| Setup | Process | KGE_Q | PBIAS_Q | KGE_N | PBIAS_N | Period_Q | Period_N |
|---|---|---|---|---|---|---|---|
| M1 | Calibration | 0.56 | 2.7 | 0.33 | −4.5 | 1985–1999 | 1993–1999 |
| | Validation | 0.57 | 8.4 | 0.11 | −30 | 2000–2015 | 2000–2008 |
| M2 | Calibration | 0.71 | 7 | 0.64 | −16 | 1981–2002 | 1994–2002 |
| | Validation | 0.71 | −12 | 0.54 | −23 | 2003–2016 | 2003–2008 |
| Zaya | Calibration | 0.51 | −7 | 0.67 | −27 | 1981–2010 | 2003–2010 |
| | Validation | 0.51 | −16 | 0.43 | −36 | 2010–2016 | 2010–2016 |

In the Zaya catchment, the simulated discharge and $NO_3^-$ were both underestimated (Figures S6 and S7). One reason may be because irrigation was not included in the model, whereas farmers in the neighboring Marchfeld region commonly practice irrigating spring barley and winter wheat during the growing season [36].

### 3.2. Evaluation of the Calibrated and Validated SWAT Model

#### 3.2.1. Water Balance

The SWAT model performed better in some years than in others, which could overall be determined by examining the annual water balance from 1985 to 2015 for each setup (M1, M2, and Zaya). Based on 31 simulation years, the percentage errors of water balance in the Melk catchment with M1 and M2 setups ranged from −5.6% to 3.0% and −11.8% to 3.6%, respectively. The percentage error of water balance in the Zaya catchment ranged from −38.7% to 21.4% (Tables S6–S8). Ideally, the long-term water balance should be 0% whereby the input and output components balance each other. Here, we use an annual water balance 0% as one additional indicator of good hydrological model performance.

#### 3.2.2. Remote Sensed and Simulated ET

Compared to remote sensed ET from the ALEXI model, the SWAT model underestimated ET for higher values (above 50 mm) in the Melk catchment, and overestimated ET for smaller values (below 25 mm) in the Melk catchment with M2 setup (Figure 3A). Overall, the SWAT model could simulate ET quite well for the Melk catchment with a relative coefficient $R^2$ for the M1 and M2 setups of 0.90 and 0.94, respectively.

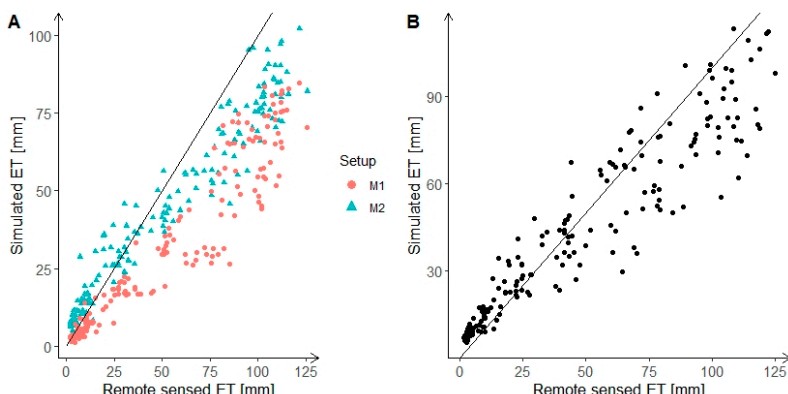

**Figure 3.** Scatter plot for simulated and remote sensed monthly ET for the years 2001–2015 in the Melk catchment (**A**) and Zaya catchment (**B**). The black line is the reference line with slope 1.

In the Zaya catchment, although the $R^2$ of the SWAT simulated ET and ALEXI remote sensed ET was 0.89, the SWAT model underestimated ET for larger values (above 50 mm) and tended to overestimate ET for smaller values (below 25 mm) (Figure 3B).

### 3.2.3. Measured and Simulated Crop Yields

The main crop yields from SWAT were compared to available measured data at the district level. Figures 4–6 show simulated crop yields in the Melk and Zaya catchments at the HRU level compared to reported yields for the years 1993–2015. For the Melk M1 setup, the spring barley yields were consistently and grossly overestimated (Figure 4). The BARL category in SWAT included several other crops that cover a wide range of yields (Table S2), for example, in addition to spring grains, the BARL category included flax, buckwheat, and quinoa. Although the SWAT simulated yields for CORN, SOYB, and WWHT varied across the HRUs, the median of simulated crop yields from all HRUs fit the measured crops yields rather well (Figures 4 and 5).

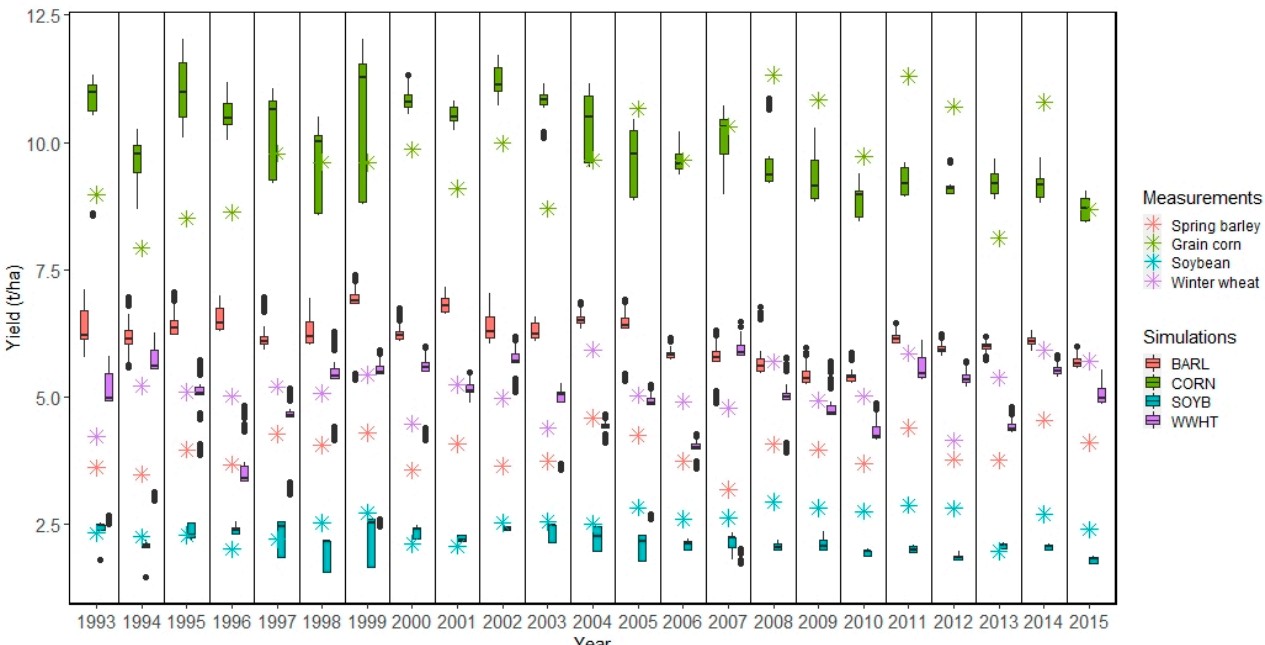

**Figure 4.** Measured and simulated crop yields at the HRU level in the Melk catchment for M1 setup. The SWAT simulated crops BARL (spring barley), CORN (grain corn), SOYB (soybean), and WWHT (winter wheat) refer to Table S2.

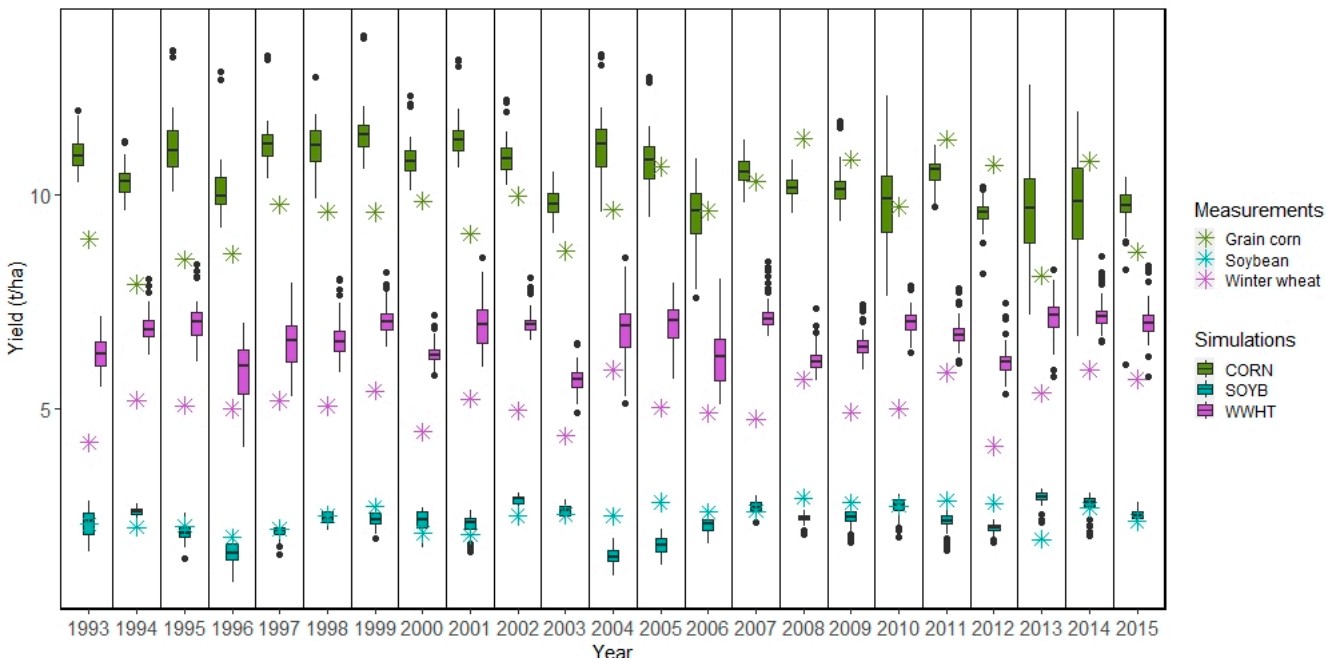

**Figure 5.** Measured and simulated crop yields at the HRU level in the Melk catchment for M2 setup. The SWAT simulated crops CORN (grain corn), SOYB (soybean), and WWHT (winter wheat) refer to Table S3.

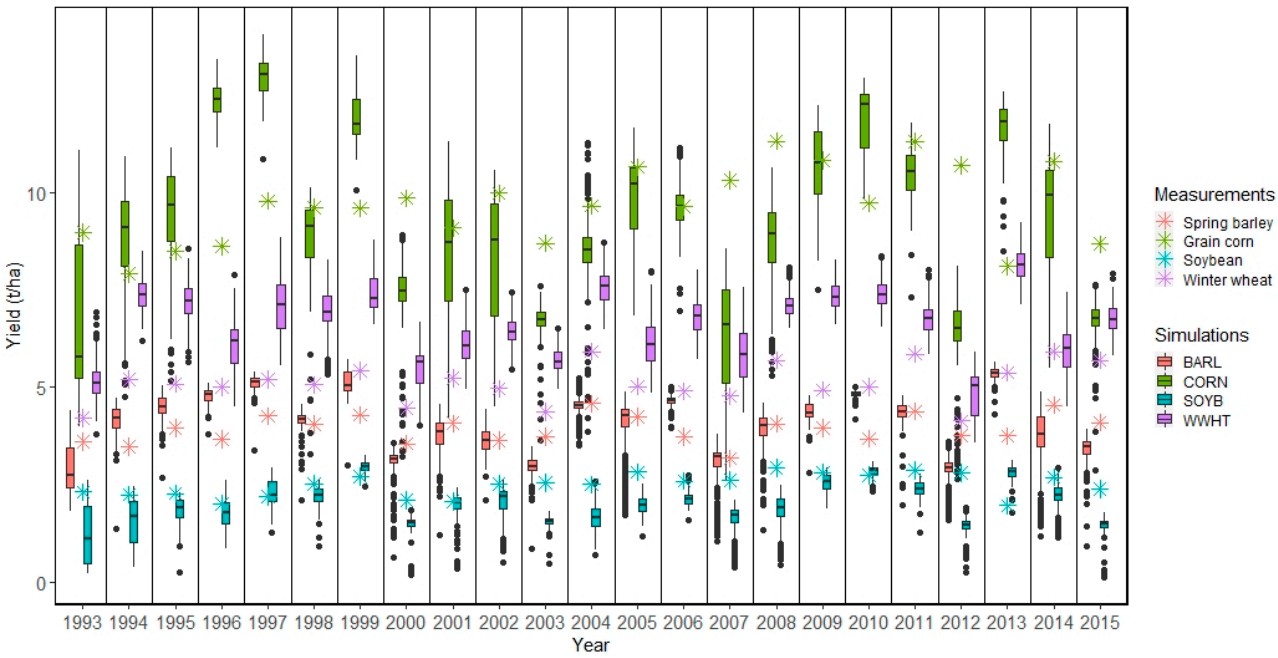

**Figure 6.** Measured and simulated crop yields at the HRU level in the Zaya catchment. The SWAT simulated crops BARL (spring barley), CORN (grain corn), SOYB (soybean), and WWHT (winter wheat) refer to Table S4.

For the Zaya catchment, not including irrigation in the SWAT model may have been a reason for some of the deviations in simulated yields especially for grain corn (Figure 6).

### 3.3. The Fraction of Nitrified N Lost as N$_2$O

N$_2$O is the by-product of the nitrification process. After the N flux from the nitrification process is simulated, the fraction (K$_2$) that partitions N$_2$O emissions from nitrification was applied to the SWAT N$_2$O submodule (Figure 2).

The fraction of nitrified N lost as N$_2$O varies with soil type (Table S1). The Melk catchment was set up with SWAT using input soil data from SoilGrids in which the soil texture for the entire catchment was simulated as one soil type, namely, silt loam (Table 1). Literature values for the fraction K$_2$ measured from a similar soil texture were collected (Table 5) and the median value of the K$_2$ was calculated to be 0.02. This value was applied as the fraction K$_2$ value for the Melk catchment. Wagena et al. [23] also set the K$_2$ value to 0.02, based on measured N$_2$O data in their study area.

**Table 5.** The fraction K$_2$ collected from literature, which has a similar soil texture with the Melk catchment.

| Reference | K$_2$ (%) | Clay (%) | Silt (%) | Sand (%) |
|---|---|---|---|---|
| Bremner and Blackmer [52] | 3.5 | 25–27 | 36–47 | 28–37 |
| Lipschultz et al. [53] | 0.15–2.5 | | Average value | |
| Goodroad and Keeney [54] | 0.1–1.1 | 19 | 50 | 31 |
| Remde and Conrad [55] | 0.1–3.9 | | Average value | |
| Garrido et al. [56] | <−0.001–1 | 20–32 | | |
| Khalil et al. [57] | 0.16–1.48 | 20 | 73 | 7 |
| Bateman and Baggs [58] | 0.17–0.53 | 15 | 68 | 17 |
| Mathieu et al. [59] | 0.13–2.32 | 14 | 52 | 35 |
| Mørkved et al. [60] | <0.1–27 | 13 | 68 | 19 |
| Mørkved et al. [61] | 0.02–7.6 | 21 | 40 | 39 |
| Frame and Casciotti [62] | 0.4–2.2 | | Average value | |

For the SWAT model setup in the Zaya catchment, the mineral N fertilizer applied did not contain NH$_4$, the simulated nitrification was zero and we did not specifically determine the fraction K$_2$ for the Zaya catchment.

### 3.4. Simulated N$_2$O Emissions and the EFs for N$_2$O

The N$_2$O submodule was applied to estimate N$_2$O emissions for the years 2006–2015 from the following dominant crops in the Melk catchment: winter wheat, grain corn, spring barley and pasture for M1 and M2 setups (Table 6). In the Zaya catchment, N$_2$O emissions from the main crops winter wheat, spring barley, grain corn and sugar beet were examined (Table 6). The ranges of simulated annual N$_2$O emissions in the Zaya catchment varied widely with crops (Table 6). The largest simulated N$_2$O emissions in the M1, M2 and Zaya setups are from grain corn, soybean and sugar beet, respectively.

### 3.5. Comparison of Simulated N$_2$O Emissions with Measured N$_2$O

We are fully aware that it is not possible to directly compare the N$_2$O emissions from different sites to each other, since the timing and absolute values of N$_2$O emissions will be different, as they depend on the local weather, soil characteristics, and field management practices. In this study, we want to compare the magnitude of emissions and the emission trends from similar crops grown on similar soil.

#### 3.5.1. The Melk Catchment

The amount and type of applied N fertilizer in Pötting, M1 and M2, are listed in Table 3. The measured N$_2$O emissions and the simulated N$_2$O emissions in 2013 from the

HRUs with grain corn in the M1 and M2 setups were plotted to examine the trends of simulated $N_2O$ emissions in the Melk catchment (Figure 7).

**Table 6.** Simulated annual $N_2O$ emissions (N kg ha$^{-1}$) and calculated EFs for $N_2O$ (%) for the crops in the Melk and Zaya catchments for the years 2006–2015.

| Setup | Variable | Function | CORN | WWHT | SOYB | PAST | BARL | SGBT |
|-------|----------|----------|------|------|------|------|------|------|
| M1 | $N_2O$ (N kg ha$^{-1}$) | Maximum | 8.59 | 11.25 | | 12 | 5.46 | |
| | | Median | 1.35 | 1.29 | | 1.03 | 1.11 | |
| | | Minimum | 0.58 | 0.67 | | 0 | 0.89 | |
| | $N_2O$ EF (%) | Maximum | 4.34 | 6.62 | | 8.57 | 4.2 | |
| | | Median | 0.68 | 0.76 | | 0.74 | 0.85 | |
| | | Minimum | 0.29 | 0.4 | | 0 | 0.68 | |
| M2 | $N_2O$ (N kg ha$^{-1}$) | Maximum | 3.95 | 2.47 | 2.56 | 0.09 | | |
| | | Median | 0.13 | 0.03 | 0.18 | 0.01 | | |
| | | Minimum | 0.03 | 0.01 | 0.04 | 0 | | |
| | $N_2O$ EF (%) | Maximum | 2.47 | 1.83 | 7.74 | 0.05 | | |
| | | Median | 0.08 | 0.02 | 0.55 | 0.006 | | |
| | | Minimum | 0.02 | 0.007 | 0.09 | 0 | | |
| Zaya | $N_2O$ (N kg ha$^{-1}$) | Maximum | 18.63 | 8.06 | | | 19.56 | 12.99 |
| | | Median | 0.85 | 0.51 | | | 0.62 | 1.94 |
| | | Minimum | 0 | 0 | | | 0 | 0 |
| | $N_2O$ EF (%) | Maximum | 12.85 | 5.58 | | | 20.06 | 11.43 |
| | | Median | 0.59 | 0.35 | | | 0.63 | 1.7 |
| | | Minimum | 0 | 0 | | | 0 | 0 |

The crops included in CORN (grain corn), WWHT (winter wheat), SOYB (soybean), PAST/FESI (pasture), BARL (spring barley), and SGBT (sugar beet) in the Melk (M1 and M2 setups) and Zaya catchments refer to Tables S2–S4, respectively.

The low spring $N_2O$ emissions from grain corn in the M1 and M2 setups in the first 120 days of the year are covered well, but the peak of $N_2O$ emissions after fertilizer application (real and simulated) are not captured by SWAT. During the remainder of the year for the M1 setup, the simulated $N_2O$ emissions from most grain corn HRUs show more peaks and are higher than the measured $N_2O$ emissions. For the M2 setup, the two main peaks of $N_2O$ emissions from grain corn were not captured by SWAT (Figure 7). The trends and timing for the low values were captured. Overall, the M2 setup had lower fluxes than the M1 setup (Figure 7).

The measured $N_2O$ emissions during the crop growing season were calculated by using linear interpolation, from these values, the average daily measured $N_2O$ emissions were calculated. Figure 8 shows the average measured daily $N_2O$ and the median, maximum and minimum values of simulated daily $N_2O$ emissions. Compared to measured daily $N_2O$ emissions, the simulated daily $N_2O$ emissions from grain corn in the M1 and M2 setups were lower by 5.37 and 12.27 N g ha$^{-1}$, respectively. However, the ranges of M1 and M2 simulated daily $N_2O$ covered the measured daily $N_2O$.

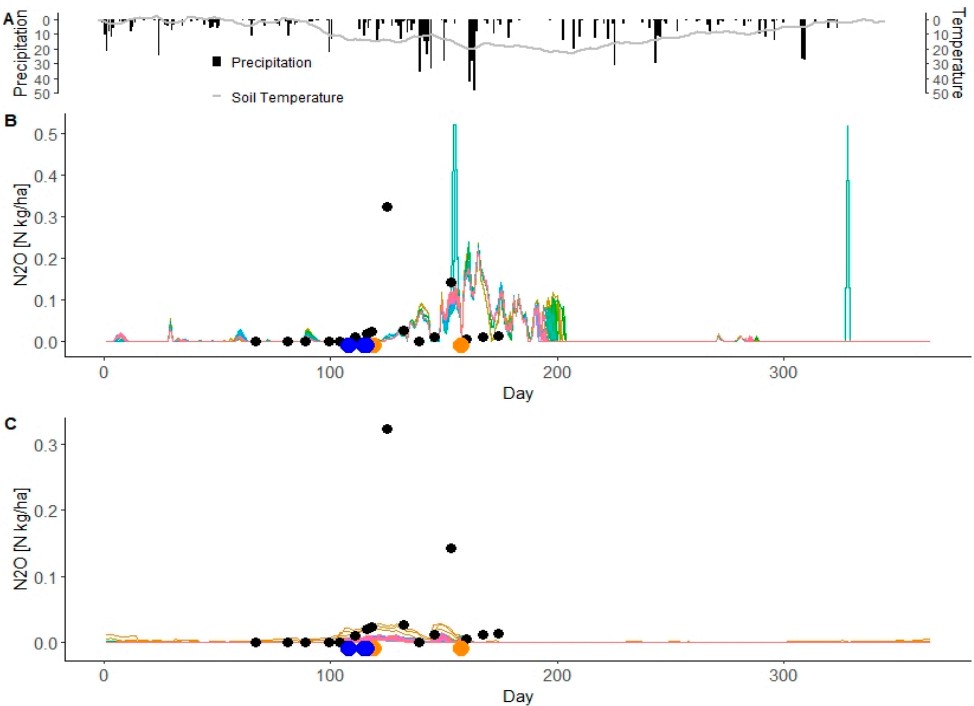

**Figure 7.** Simulated N$_2$O emissions from grain corn in the Melk catchment at the HRU level (color lines) and measured N$_2$O emissions (black points) from grain corn in Pötting in 2013 (**A**). (**B**,**C**) are M1 setup and M2 setup, respectively. Blue points indicate the dates of N fertilizer application in Pötting and orange points show the N fertilization in the SWAT simulations.

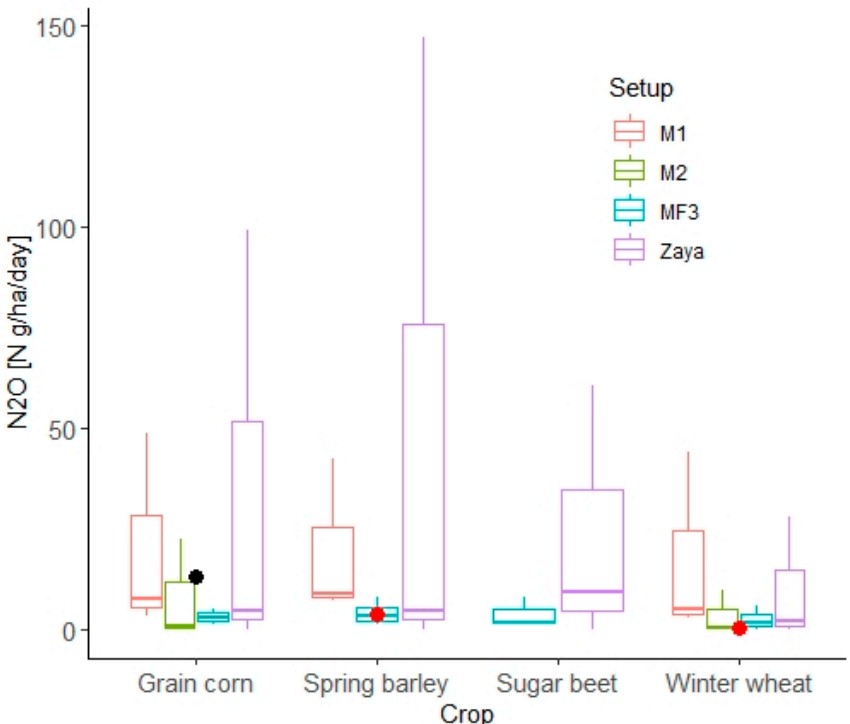

**Figure 8.** The comparison of average simulated daily N$_2$O and average measured daily N$_2$O during the growing season. The boxplot shows the median with hinges on the 25% and 75% quantiles. Black point indicates measured average daily N$_2$O in Pötting. Red points indicate measured average daily N$_2$O in Hirschstetten.

### 3.5.2. The Zaya Catchment

In Hirschstetten in 2018, the measured $N_2O$ emissions from spring barley were obtained after it was planted, after the N fertilizer was applied and after a heavy rainfall [63]. The $N_2O$ emissions from winter wheat were measured in 2019 after N fertilizer was applied and after the harvest [64]. Precipitation was applied during the growing season to spring barley 346 mm and to winter wheat 352 mm [63,64].

From the SWAT simulations, two years (2009 and 2013) with a good performing model (i.e., when the absolute water balance was <11%) and two years (2014 and 2015) with bad performing model (i.e., when the absolute water balance was >11%) were selected to check the daily $N_2O$ emissions from spring and winter wheat. For the years 2009 and 2013 the percentage errors of simulated water balance are 8.1% and 0.0%, respectively. The simulated $N_2O$ emissions occurred mainly during the crop growing season, after N fertilizer application, although the impact of N fertilizer application on $N_2O$ emissions was clearer for spring barley (Figure 9A,B and Figure 10A,B). For two years in which the simulated water balance was poor (in 2014 and 2015 with 21.4% and −23.8%, respectively), the impact of N fertilizer application on $N_2O$ emissions could not be captured (Figure 9C,D and Figure 10C,D).

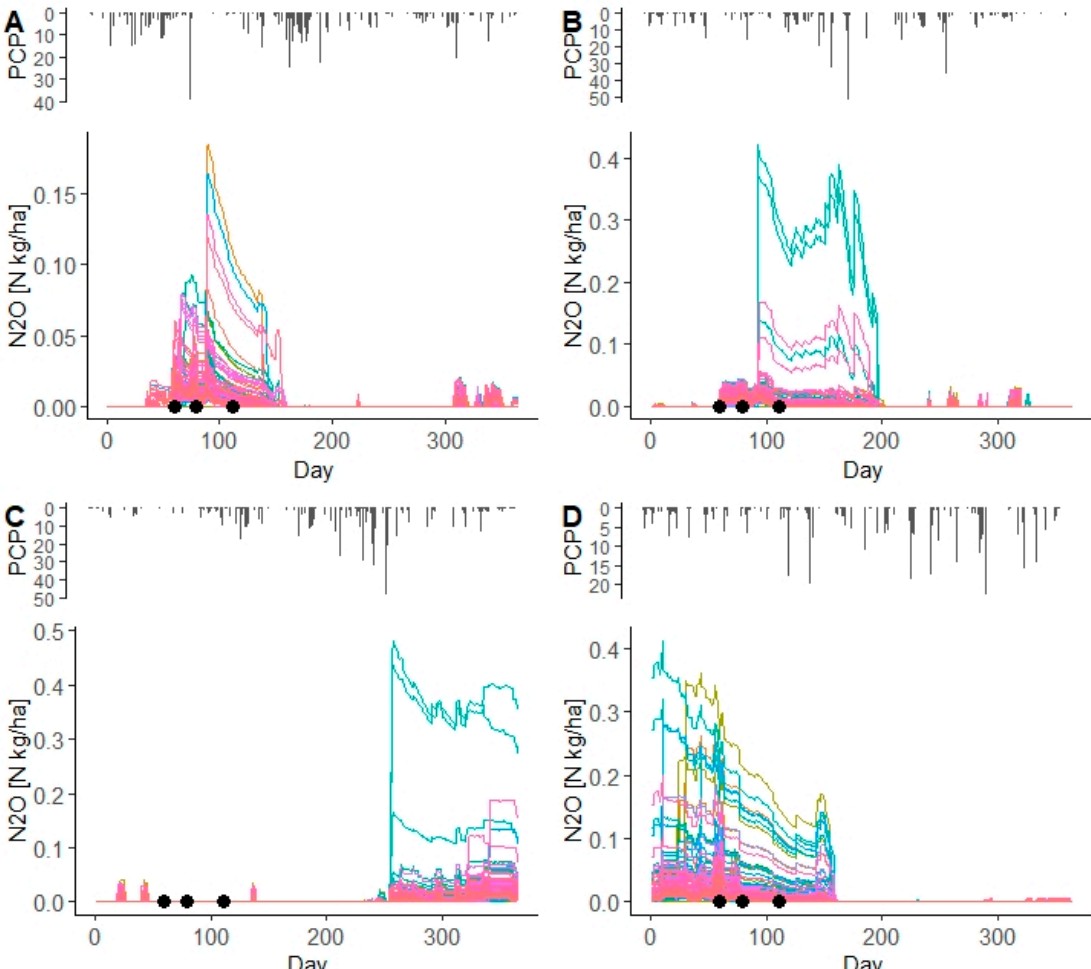

**Figure 9.** Simulated $N_2O$ emissions (color lines) from BARL HRUs in the Zaya catchment for the years 2009 (**A**), 2013 (**B**), 2014 (**C**), and 2015 (**D**). PCP is daily precipitation (mm). Black points indicate the dates of N fertilizer application for the setup in the Zaya catchment.

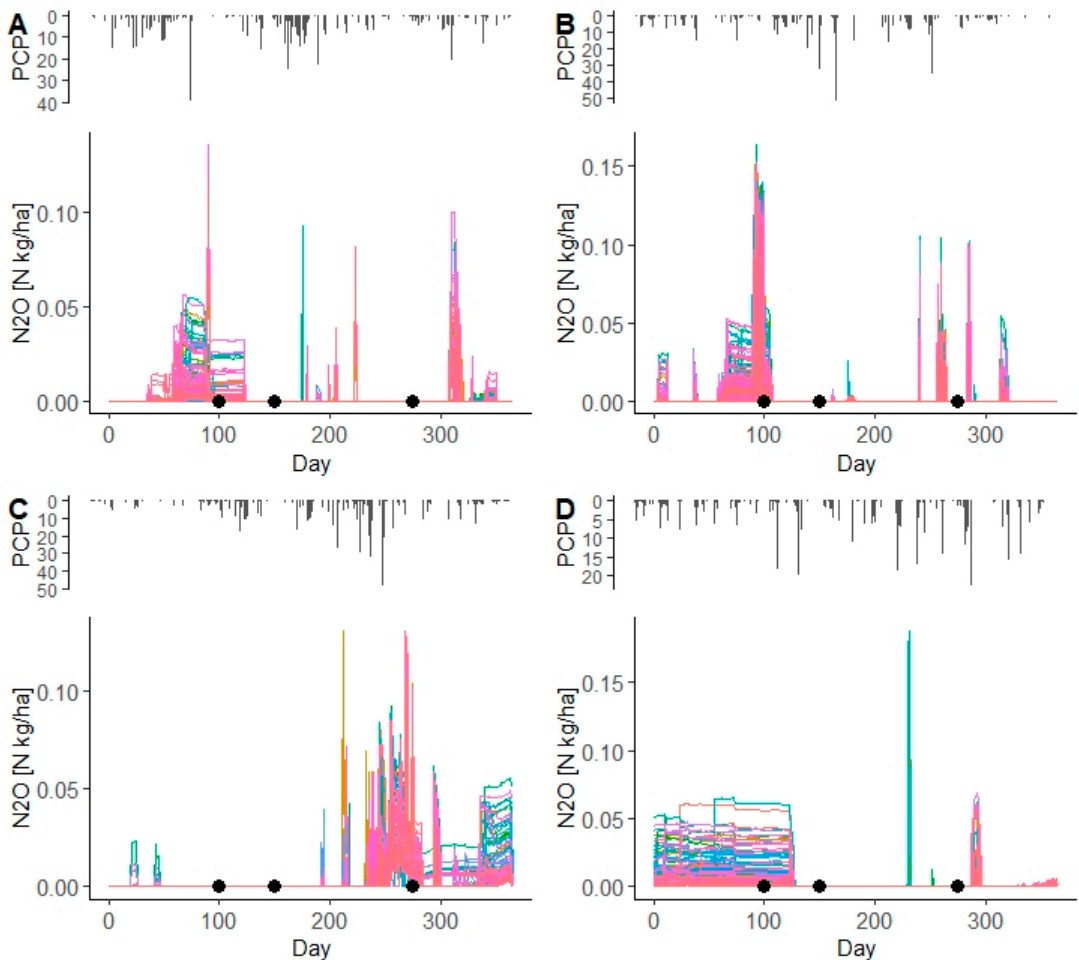

**Figure 10.** Simulated N$_2$O emissions (color lines) from WWHT HRUs in the Zaya catchment for the years 2009 (**A**), 2013 (**B**), 2014 (**C**), and 2015 (**D**). PCP is daily precipitation (mm). Black points indicate the dates of N fertilizer application for the setup in the Zaya catchment.

The simulated average daily N$_2$O from spring barley and winter wheat in the Zaya catchment were 0.66 and 1.6 N g ha$^{-1}$ higher, respectively compared to the average measured daily N$_2$O emissions from spring barley and winter wheat in Hirschstetten (calculated based on linear interpolation) (Figure 8).

*3.6. Comparing Simulated N$_2$O Emissions with DNDC Simulations*

For the M1 setup in the Melk catchment, the same type of N fertilizer, namely, NH$_4$NO$_3$, was also applied in the DNDC model for MF3 in Kasper et al. [36] (Table 3). The SWAT simulated daily N$_2$O from grain corn, winter wheat, and spring barley were higher by 4.73, 3.54, and 5.5 N g ha$^{-1}$, respectively, compared with the DNDC simulated N$_2$O in MF3 (Figure 8). For the M2 setup, the applied N fertilizer for grain corn and winter wheat were higher by 10.3 and 18.8 kg ha$^{-1}$, respectively, than the N fertilizer application in MF3 (Table 3). However, the median values of simulated average daily N$_2$O emissions from grain corn and winter wheat were lower by 2.17 and 1.39 N g ha$^{-1}$, respectively, than the simulated N$_2$O emissions in MF3 (Figure 8).

In the Zaya catchment, the simulated daily N$_2$O emissions from grain corn, winter wheat, spring barley, and sugar beet were higher by 1.62, 0.26, 1.56, and 7.32 N g ha$^{-1}$, respectively, than simulated daily N$_2$O emissions in MF3, regardless of the amount of fertilizer applied (Figure 8).

### 3.7. EFs for N$_2$O for the Melk and Zaya Catchments

Using the 31 years of climate data in the simulation period, the ratio of annual precipitation/potential evapotranspiration is calculated to be 1.18 for the Melk catchment and 0.66 for the Zaya catchment. Melk is considered to be a wet climate, and Zaya is in a dry climate zone. According to IPCC, the description "mineral N" includes both the category of mineral N as well as the category of a fertilizer mixture, which includes both mineral and organic N [5]. The simulated N$_2$O EFs from the Melk catchment were compared with the aggregated IPCC default N$_2$O EF value (1%) and disaggregated IPCC default N$_2$O EF for wet climate applied with mineral N (1.6%) (Figure 11A). The simulated N$_2$O EFs from the Zaya catchment were compared with the aggregated IPCC default N$_2$O EF value (1%) and disaggregated IPCC default N$_2$O EF for dry climate (0.5%) (Figure 11B).

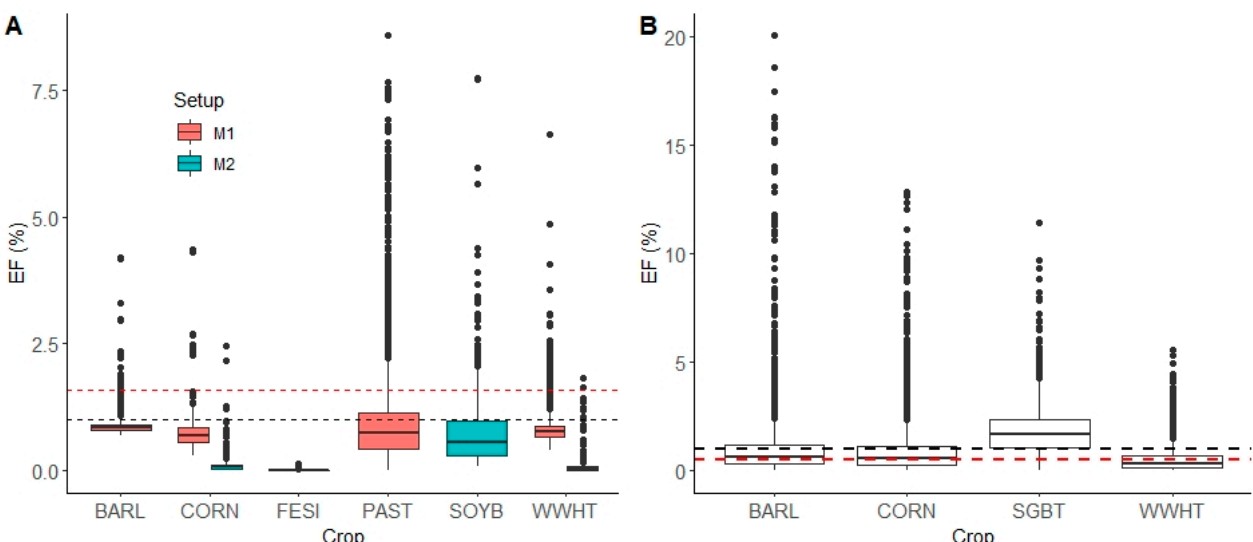

**Figure 11.** Boxplots of the simulated N$_2$O EFs from the main crops in the Melk (**A**) and Zaya (**B**) catchments for all HRUs for 2006–2015. The boxplot shows the median with hinges on the 25% and 75% quantiles. The black dashed line indicates the aggregated IPCC default N$_2$O EF value 1%. The red dashed line in (**A**) indicates the disaggregated IPCC default N$_2$O EF for wet climate (1.6%) and in (**B**) indicates disaggregated IPCC default N$_2$O EF value 0.5% for dry climate. The crops included in BARL (spring barley), CORN (grain corn), FESI (intensively managed pasture), PAST (pasture), SOYB (soybean), WWHT (winter wheat), and SGBT (sugar beet) are listed in Tables S2–S4.

The median simulated N$_2$O EFs in the Melk catchment in the M1 and M2 setups are lower than the aggregated and disaggregated IPCC default values (Figure 11A). Particularly, the N$_2$O EFs with M2 setup are much lower than the N$_2$O EFs. The median simulated N$_2$O EFs in the Zaya catchment for BARL, CORN and WWHT were between the aggregated and disaggregated IPCC values and showed fairly good agreement with the IPCC calculations. However, the outliers for some HRUs and years showed much higher values, with EFs up to 20%.

## 4. Discussion

### 4.1. Model Performance for Simulating N$_2$O Emissions

In the M1 setup, the SWAT simulated average daily N$_2$O during the growing season from grain corn was 70% lower than the measured daily N$_2$O in Pötting (Figure 8). Aggregating measured hourly N$_2$O to the daily level during the growing season in Pötting and the linear interpolation method both tend to overestimate the N$_2$O emissions [48,65]. In addition, the M1 simulated N$_2$O showed more frequent peaks than the measured N$_2$O data in Pötting (Figure 7). The reasons are (1) more than 35 kg ha$^{-1}$ N fertilizer was applied to the grain corn in the M1 setup, which contributes to more N$_2$O emissions (Table 3,

Figure 8). (2) The dates of N fertilizer application in the M1 setup were split according to different crop growing stages (Figure 7), which influenced the efficiency of fertilizer use and $N_2O$ emissions [66]. (3) The samples of $N_2O$ measurements were very few, which leads to the missing of peaks of $N_2O$ emissions [8].

As well, the average daily $N_2O$ emissions from grain corn, winter wheat, and spring barley simulated with the M1 setup were 62%, 70%, and 64% higher, respectively, compared to simulated $N_2O$ emissions in Kasper et al. [36] (Figure 8). The reasons are (1) the N fertilizer application for grain corn, winter wheat and spring barley in the M1 setup were 47.6, 54, and 75 kg ha$^{-1}$, respectively, higher than MF3 (Table 3). (2) During the comparison period, the Melk catchment had an average of 85 mm more precipitation annually than the area where DNDC was simulating $N_2O$. Therefore, the $N_2O$ emissions were correspondingly higher in the Melk during this period. (Table 3) [67].

In the M2 setup, the simulated average daily $N_2O$ emissions from grain corn were 12.27 and 2.17 N g ha$^{-1}$ lower than measured daily $N_2O$ emissions in Pötting and simulated daily $N_2O$ in MF3, respectively. The M2 simulated daily $N_2O$ from winter wheat was 1.39 N g ha$^{-1}$ lower than MF3 simulated by DNDC (Figure 8). The type of N fertilizer used as input in the M2 setup influenced the simulated N flux in the SWAT model. In the M2 setup, the N fertilizer is a combination of organic N and mineral fertilizer ($NO_3^-$). In the SWAT model, the organic N is added to the soil $NO_3^-$ through the mineralization process, and then $NO_3^-$ is reduced to $N_2O$ or $N_2$ by denitrification [29]. In the M2 setup, the $N_2O$ emissions occur, therefore, only during the denitrification process. The denitrification parameter (CDN) was set at a very low value in the M2 setup during the calibration to meet the in-stream $NO_3^-$ concentration data (Table S5), as a result, low denitrification, $N_2O$ emissions and $N_2O$ EFs were simulated (Figures 8 and 11A).

For soybean in the M1 setup, no N fertilizer was applied to this crop; however, in the M2 setup, N fertilizer was applied to soybean (33 kg ha$^{-1}$), which caused high simulated $N_2O$ emissions. Overall, they were the highest in soybean compared to grain corn and winter wheat. However, the soybean yields were not significantly improved with N fertilizer application in the SWAT model.

For both M1 and M2 setups, the simulated $N_2O$ emissions mainly occurred during the crop growing season and after N fertilizer application, which fits the findings based on measured $N_2O$ data [35].

In the Zaya setup, the simulated average daily $N_2O$ emissions were higher than measured daily $N_2O$ in Hirschstetten (16.5% and 1000% higher for spring barley and winter wheat, respectively) (Figure 8). Simulated daily $N_2O$ for grain corn, winter wheat, spring barley and sugar beet were higher by 56%, 17%, 50%, and 430%, respectively, than MF3 simulated by DNDC (Figure 8).

The type of N fertilizer for the Zaya setup is the same as the M2 setup, and simulated $N_2O$ emissions occurred only by denitrification. The denitrification parameter CDN was also set at a low value (Table S5). However, the simulated $N_2O$ emissions in the Zaya catchment were higher than the M2 setup (3.79 and 1.65 N g ha$^{-1}$ day$^{-1}$ higher for grain corn and winter wheat, respectively). The M2 setup had more N fertilizer application, silt loam soil as input and more than 393 mm annual precipitation (Table 3). The positive relationships between $N_2O$ emissions and N fertilizer application, soil texture and precipitation had been described previously, for example, Wang et al. [12].

As Figures 9 and 10 show, the impact of N fertilizer application on simulated $N_2O$ emissions from spring barley and winter wheat in the Zaya catchment was captured for the years 2009 and 2013, which were years when the simulated water balance was <11%.

Precipitation influences $N_2O$ emissions in the Zaya catchment by influencing soil moisture and further affecting denitrification and the $N_2/N_2O$ ratio. Based on the ANOVA analysis for the years 2006–2015, the significant impact of daily precipitation on simulated daily $N_2O$ emissions from spring barley in the Zaya catchment can be identified for the years 2006, 2009, 2010, and 2013, when the absolute percentage error of the simulated water balance were good (−11.4%, 8.1%, 10.4%, and 0.0%, respectively) (Tables S8 and S9).

### 4.2. Uncertainties of Simulated $N_2O$ Emissions

To quantify the uncertainties in the simulated $N_2O$ emissions, we varied the factors that influence the simulated $N_2O$ emissions in the submodule, such as the fraction $K_2$, simulated soil $NO_3^-$, SOC, and soil water. These factors were changed by $\pm10\%$, $\pm20\%$, and $\pm30\%$ to analyze the impact of each factor on $N_2O$ emissions in winter wheat (Figure 12) and grain corn (Figure 13) in the Melk catchment with M1 and M2 setups. The same analysis was performed in the Zaya catchment for winter wheat and spring barley.

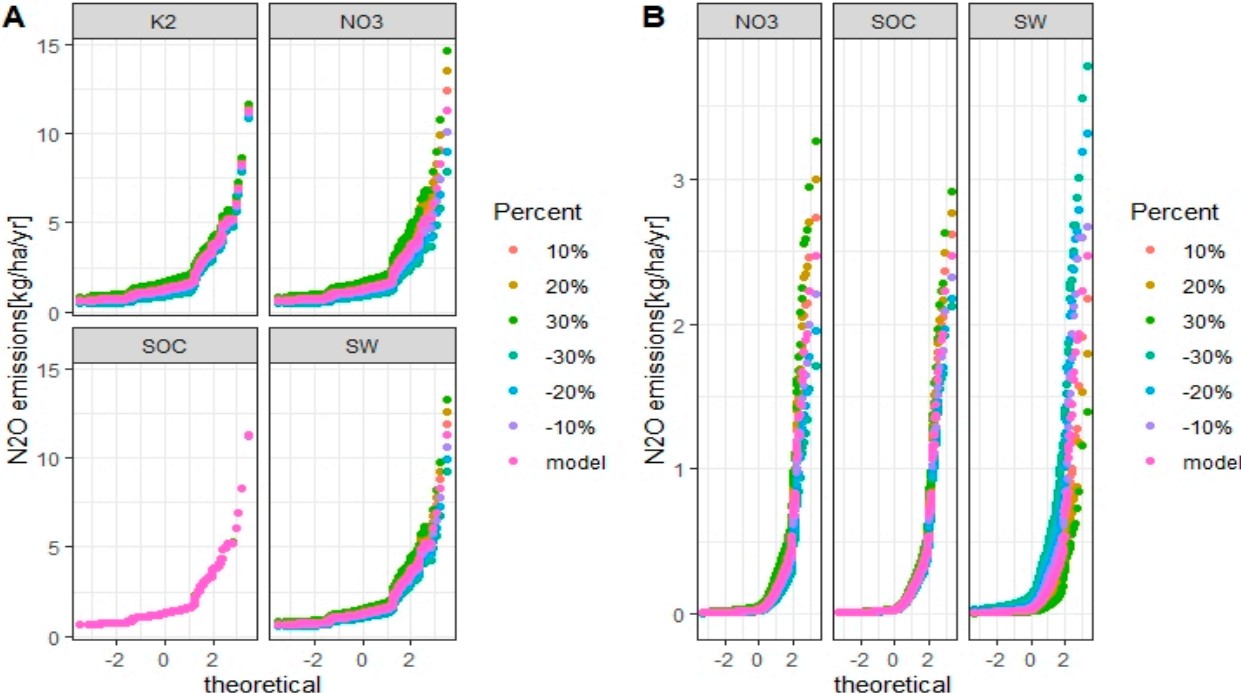

**Figure 12.** Uncertainties of simulated annual $N_2O$ emissions from winter wheat in the Melk catchment with M1 (**A**) and M2 (**B**) setups for each considered factor. The X-axis is the standard normal distribution of simulated $N_2O$ emissions with mean 0 and standard deviation 1. Model is the validated SWAT model and is the baseline.

For the M1 setup, the fraction $K_2$, soil $NO_3^-$, and soil water are sensitive factors that influence the simulation of $N_2O$ emissions (Figures 12A and 13A). For the M2 setup, soil $NO_3^-$ and soil water are sensitive factors that influence the simulation of $N_2O$ emissions (Figures 12B and 13B).

When the factors were changed by +30%, the simulated $N_2O$ emissions from winter wheat and grain corn increased by 99% and 98% for M1 setup, 93% and 79% for M2 setup, respectively, compared to baseline. With −30% of factors, the simulated $N_2O$ emissions from winter wheat and grain corn compared to baseline decreased by 60% and 60% for M1 setup, 30% and 27% for M2 setup, respectively.

For the Zaya catchment, soil $NO_3^-$ and soil water are sensitive factors that influence the simulation of $N_2O$ emissions from winter wheat. Soil $NO_3^-$, SOC, and soil water are sensitive factors that influence the simulation of $N_2O$ emissions from spring barley (Figure 14).

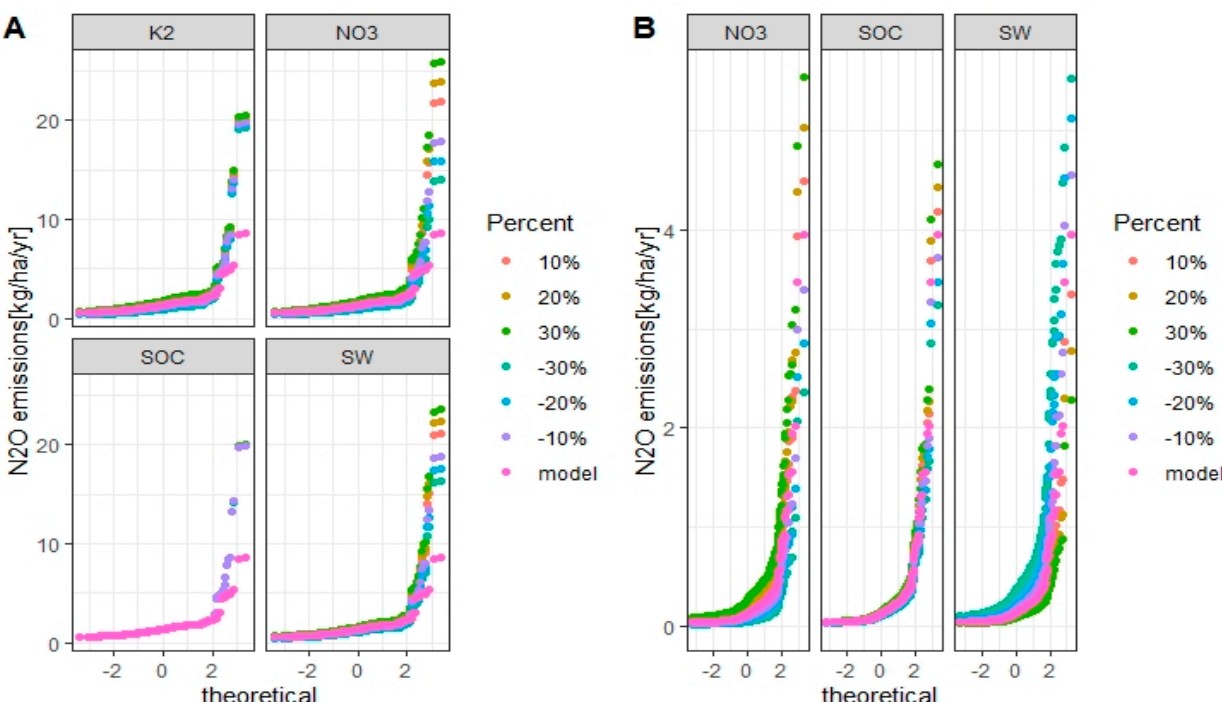

**Figure 13.** Uncertainties of simulated annual N₂O emissions from grain corn in the Melk catchment with M1 (**A**) and M2 (**B**) setups for each considered factor. The X-axis is the standard normal distribution of simulated N₂O emissions with mean 0 and standard deviation 1. Model is the validated SWAT model and is the baseline.

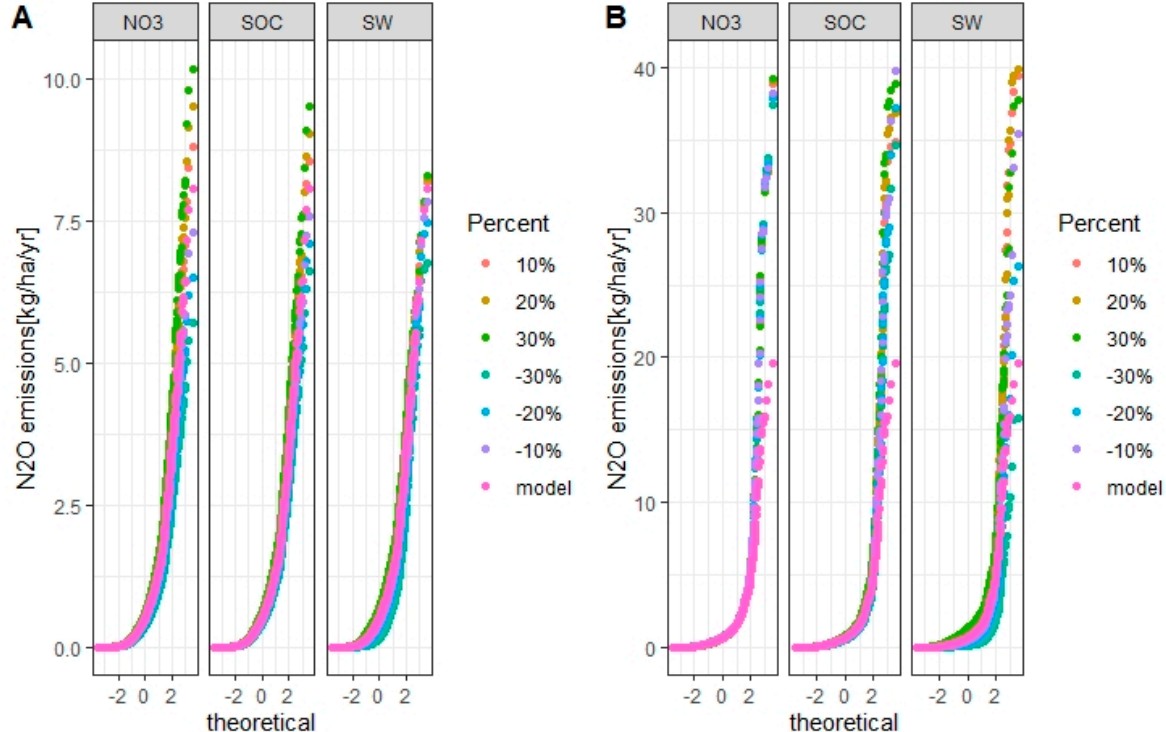

**Figure 14.** Uncertainties of simulated annual N₂O emissions from winter wheat (**A**) and spring barley (**B**) in the Zaya catchment for each considered factor. The X-axis is the standard normal distribution of simulated N₂O emissions with mean 0 and standard deviation 1. Model is the validated SWAT model and is the baseline.

In the Zaya catchment, when the values were changed by +30%, the simulated $N_2O$ emissions from winter wheat and spring barley increased by 88% and 135%, respectively, compared to baseline. When the values were decreased by 30%, the simulated $N_2O$ emissions from winter wheat and spring barley decreased by 80% and 86%, respectively, compared to baseline.

To highlight future areas for model improvement, the uncertainties are listed to report on the limitations of the simulated variables (Table 7).

**Table 7.** Sources of uncertainty for simulating $N_2O$ emissions with the $N_2O$ submodule in the Melk and Zaya catchments.

| Sources of Uncertainty | Description |
|---|---|
| The SWAT model | 1. The SWAT model includes a number of parameters. The measured values for parameters in Table S5 are not available. |
| | 2. The SWAT model cannot account for the changes in soil pH, which is influenced by the amount of rainfall, the soil texture, crop residue, commercial fertilizer application, and different management practices. |
| Input data | 1. The soil data from SoilGrids and European soil data center have low resolution 250 m and 1 km, respectively. |
| | 2. For the land use map, we grouped in the INSPIRE databased into similar categories for SWAT. The simulated $N_2O$ emissions were therefore not only from one crop type, but from crops included within each group. |
| | 3. Irrigation data was not available. |
| The fraction $K_2$ | The fraction $K_2$ used to partition $N_2O$ emissions from nitrification was not measured in the Melk catchment, bur collected from literature. The fraction $K_2$ varies with soil texture and climate. |
| Semi-empirical equations | The semi-empirical equations used in this study to calculate the ratio $N_2/N_2O$ are from Parton et al. [26], which were developed based on laboratory data and the data in Weier et al. [31]. In fact, the impacts of soil $NO_3^-$, SOC, and soil moisture on the $N_2/N_2O$ ratio are highly variable spatially. |
| Measured $N_2O$ data | For the Melk catchment and Zaya catchment, no measured $N_2O$ data is available. We have used available measured data from proximate regions with similar crops, soil types, and climates. |
| $NO_x$ | $NO_x$ is also the obligate intermediate in the denitrification process. In this study, we did not partition $NO_x$ from the denitrification process, which may overestimate $N_2O$ emissions. |

## 5. Conclusions

In this study, we developed an $N_2O$ submodule for the SWAT model to improve the ability for simulating $N_2O$ emissions based on spatially distributed hydrological and N outputs at the catchment scale. The $N_2O$ submodule was developed by partitioning $N_2O$ from SWAT simulated nitrification with the fraction $K_2$ and also by isolating $N_2O$ from SWAT simulated denitrification ($N_2O + N_2$) using a modified semi-empirical equation. The $N_2O$ emissions and $N_2O$ EFs from the main crops (e.g., grain corn, winter wheat, spring barley, and soybean) in the Melk and Zaya catchments were quantified from the developed $N_2O$ submodule.

By comparing $NH_4NO_3$ as N fertilizer input data with a combination of organic N and $NO_3^-$ as fertilizer input, we found the type of N fertilizer chosen for the SWAT model influences the amounts of simulated $N_2O$ emissions. In the Melk catchment, the setup with the combination of organic N and $NO_3^-$ fertilizer simulated lower $N_2O$ emissions from grain corn and winter wheat than the setup with $NH_4NO_3$ fertilizer. The simulated $N_2O$

emissions in the Melk setup with $NH_4NO_3$ fertilizer were closer to the measured $N_2O$ data than the other model setups.

The SWAT model performance for simulating hydrological processes influences the simulation of N fluxes and $N_2O$ emissions. In the Zaya catchment, by examining selected years in which the absolute percentage error of the simulated water balance was good (<11%), we found daily precipitation significantly impacts $N_2O$ emissions, and the $N_2O$ submodule captured the impact of N fertilizer application on $N_2O$ emissions for both catchments, and was in agreement with the existing measured $N_2O$ data. The hot moments of simulated $N_2O$ emissions were during the crop growing season and after N fertilizer application.

The $N_2O$ submodule developed in this study should be further tested in catchments with existing measured $N_2O$ data and under a wider range of crops and climates.

**Supplementary Materials:** The following supporting information can be downloaded at: https://www.mdpi.com/article/10.3390/atmos13010050/s1, Figure S1. The representation of the equations for the impacts of soil $NO_3^-$ (A) and soil water (B) on the ratio of $N_2/N_2O$ in Parton et al. [26] and Wagena et al. [23]. Figures S2–S7. Hydrographs for discharge and $NO_3$. Table S1. Literature values of the fraction of nitrified N lost as $N_2O$, which is denoted by $K_2$. Tables S2–S4. Crop classification into groups for the M1, M2 and Zaya setups, respectively. Table S5. The calibrated parameters with their initial ranges and final validated values for the Melk and Zaya catchments. Tables S6–S8. SWAT simulated water balance for the M1, M2 and Zaya setups, respectively. Table S9. ANOVA analysis for identifying the impact of daily precipitation on daily $N_2O$ emissions from spring barley. References [68–85] are cited in the Supplementary Materials.

**Author Contributions:** Conceptualization, C.W. and B.M.-S.; methodology, C.W., B.M.-S., and K.S.; software, C.W. and C.S.; formal analysis, C.W.; writing—original draft preparation, C.W.; writing—review and editing, B.M.-S. and K.S.; measured $N_2O$ data in Hirschstetten, A.W.; measured $N_2O$ data in Pötting, G.B.; fertilizer data for M2 and Zaya, O.Z. and M.Z. All authors have read and agreed to the published version of the manuscript.

**Funding:** This work is supported by the China Scholarship Council [grant numbers 201708620181] and by the NitroClimAT project [KR17AC0K13625] funded by the Austrian Climate and Energy Fund in the 10th call of the ACRP program.

**Institutional Review Board Statement:** Not applicable.

**Informed Consent Statement:** Not applicable.

**Data Availability Statement:** All input data used for setting up the SWAT model are available at: https://github.com/snailslowrun/N2O-submodule/blob/main/Input_data.docx (accessed on 19 December 2018). R codes for the $N_2O$ submodule are available at: https://github.com/snailslowrun/N2O-submodule (accessed on 19 December 2018).

**Acknowledgments:** This work was supported by the Doctoral School "Human River Systems in the 21st Century (HR21)" of the University of Natural Resources and Life Sciences, Vienna. $N_2O$ data acquisition was gained within the project Climagrocycle [KR16AC0K13275] funded in the 9th call of the ACRP program. We would like to thank Omar Chebib and Robert Michałowski for the help in modifying and compiling SWAT source codes. We would also like to thank Claire Brenner and Martha Anderson for the sharing of remote sensed ET from ALEXI model.

**Conflicts of Interest:** The authors declare no conflict of interest.

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
