# Peer review of "N2O Emissions from Two Austrian Agricultural Catchments Simulated with an N2O Submodule Developed for the SWAT Model"

_atmosphere, doi:10.3390/atmos13010050_

Round 1

Reviewer 1 Report

The manuscript “N2O emissions from two Austrian agricultural catchments simulated with an N2O submodule developed for the SWAT model” presented a new model approach to calculate the contribution of the nitrification and denitrification in the estimated N2O flux by SWAT model. The authors used two case study catchments for the model calibration and validation and used the model to compare simulated N2O flux with existing N2O data.

The topic is within the scope of the journal.

Several similar studies (collected and well cited by the authors in the introduction) tried to calculate the N2O ratio of the calculated N gas fluxes of the SWAT model and determine the source of the calculated N2O flux, however, these attempts were not always successful. The manuscript, therefore, deals with an interesting and important topic.

General comments

The manuscript is very long and difficult to read. There are a large number of tables and figures (in total 18 figures and 7 tables in the text and 7 figures and 10 tables in the supplementary material). Some parts are redundant. All sections have to be revised to different degrees. In my opinion, the manuscript cannot be published in the present form, major revision is needed.

  • The text contains several redundant definitions and long, detailed descriptions of methods, usually after proper citations. There are some examples in the specific comments.
  • The structure of the sections is inconsistent, therefore needs to be revised.
  • The tables of input data, far too detailed. Statistical descriptions and methods of data collection are unnecessary. Author could shorten the manuscript, if the model specific tables (input data, descriptions of the data) were moved to an online repository (for instance Github)
  • “Materials and Methods” content in Results.
  • Too much information! Author doesn’t need to show all the results.
  • The presented information would be enough minimum for two papers.
  • Please, avoid the application of the “therefore”, “nevertheless”, etc. worlds in the “Results” section. These expressions can be used in the “Discussion” section.

Specific comments

Line 36-41: “The global warming potential (GWP) of a GHG…” Redundant information.

Line 73-74: Missing references

Line 74-75. “Several papers...” Redundant information.

Line 171-174: Missing hypothesis

Line 185: 63%+12%, 69%+8%. What about the rest? Are there any info in the manuscript about the rest? Please, add to the manuscript.

Line 195: Table 1: Dominant texture: please add resolution (1km x 1km? or 100km x 100km? or something else?)

Line 231: Either use everywhere superscript or use “/”, but don’t use both.

Line 320: "The ÖPUL program..." Redundant information. Generally, there are a lot of unnecessary information. These information make the text long and difficult to read. The important information is lost among the redundant information.

Line 334: Table 2: 2nd and 3rd lines: Please use the proper format

Line 355-358: Redundant information

Line 362: warm-up or spin-up?

Line 515, 519, 523: Fig. 5, 6, 7:  Here and everywhere, the readers have to understand the description of the figures or tables without any extra explanation. The abbreviations (for example HRU) not acceptable without definitions.

Line 592: The resolution of the Fig. 11 is very low. It is difficult to compare the measured and modeled fluxes. I would not use the blue and orange dots. To color the area of the diagram to show the actual treatment would be better.

Line 635, 638: Figure 12, 13 Really hard to see the orange dots.

Reviewer 2 Report

This is the revision of the manuscript number atmosphere-1489208 Title: “N2O emissions from two Austrian agricultural catchments simulated with an N2O submodule developed for the SWAT model”, proposed by Miss Ms. Réka Kirmajer and colleagues for consideration for publication in Atmosphere

The manuscript raises a novel issue and attempts to address an increasingly problematic issue, the pollution of the atmosphere through emissions corresponding to agricultural soils by N2O in a given space, the ideas proposed on the models used leave a lot of information vital to understand generation and emission, so analytically this study, which was raised in two large and specific areas of Austria, leaves much to be desired, areas so large that they make it difficult to diagnose the SOC content through cartography, because the scales of the maps Used are very wide and the variety of soils makes the determination of parameters such as pH, Electric Conductivity or texture complex, these parameters are so influential for nitrification-denitrification, as well as the enzymatic activity of the soils, specifically urease activity. which is responsible for catalyzing the hydrolysis reaction of Nitrogen, the main nitrogen cell product of the degradation of proteins and nucleic acids.

I also observe inequality in the use of units, for example ug N g-1 or kg N/ha (I advise the first option).

In Table 2 it is not clear to me the values of M1 and M2 due to repeated values and cultures.

General comments:

The author reports knowledge about the models used to quantify the amount of N2O produced by nitrification-denitrification, he could monitor the influential parameters in the laboratory for this activity, knowing the cost of the approach, it could start with smaller catchments in size with what Which accessible to research, where to have controlled the greatest number of factors to give some conclusive conclusions.

Reviewer 3 Report

The carefully edited article contains interesting research results and increases the scope of knowledge. However, it requires some refinement (correction). SWAT should be full meaning

  1. The abstract should be revised considering the results and discussion of your study. I do not think that the abstract contains the essence of your research.
  2. The introduction part is too long, should be reduced.
  3. Describe the research purpose more clearly in the Introduction.
  4. The problem statement is not properly written. There should be a sequence according to 5 points-

Previous study,

Gaps in literature,

Challenges and overcome,

Significant (contribution for friendly environment/food security) and

Novelty.

  1. Line 96: DAYCENT model (full meaning)
  2. Line 98: DNDC (full meaning)
  3. Line 252: add ‘was’ before developed.
  4. Line: 282: Soil bulk density g cm-3
  5. Line 345: add ‘were’ before chosen.
  6. Line 412: add ‘are’ before similar.
  7. Line: 444, 702, 709, Table 6: ha-1
  8. The conclusion part is too long, should be reduced according to these 3 points-

Findings,

limitations, and

recommendation

  1. Some sentences are too long to read or understand. Please make it simple by splitting.
  2. All the references mentioned in the text must be included in the references and vice-versa. Unused references must be deleted. All references should be in the same style.

Line: 897, 952, 969, 978, 998 all first letter capital

Line: 1053 Soil Research

  1. Some English grammatical errors occurred. Please check the English expression and style carefully.

Round 2

Reviewer 2 Report

I accept the changes made

Reviewer 3 Report

can check spelling.